# Inconsistency, Instability, and Generalization Gap of Deep Neural Network Training

**Rie Johnson**
RJ Research Consulting
New York, USA
riejohnson@gmail.com

**Tong Zhang**[*]
HKUST
Hong Kong
tozhang@tongzhang-ml.org

## Abstract

As deep neural networks are highly expressive, it is important to find solutions with small *generalization gap* (the difference between the performance on the training data and unseen data). Focusing on the stochastic nature of training, we first present a theoretical analysis in which the bound of generalization gap depends on what we call *inconsistency* and *instability* of model outputs, which can be estimated on unlabeled data. Our empirical study based on this analysis shows that instability and inconsistency are strongly predictive of generalization gap in various settings. In particular, our finding indicates that inconsistency is a more reliable indicator of generalization gap than the *sharpness* of the loss landscape. Furthermore, we show that algorithmic reduction of inconsistency leads to superior performance. The results also provide a theoretical basis for existing methods such as co-distillation and ensemble.

## 1 Introduction

As deep neural networks are highly expressive, the *generalization gap* (the difference between the performance on the training data and unseen data) can be a serious issue. There has been intensive effort to improve generalization, concerning, for example, network architectures [15, 11], the training objective [10], strong data augmentation and mixing [6, 41, 39]. In particular, there has been extensive effort to understand the connection between generalization and the *sharpness* of the loss landscape surrounding the model [14, 20, 16, 18, 10]. [18] conducted large-scale experiments with a number of metrics and found that the sharpness-based metrics predicted the generalization gap best. [10] has shown that the sharpness measured by the maximum loss difference around the model ($m$-*sharpness*) correlates well to the generalization gap, which justifies their proposed method *sharpness-aware minimization (SAM)*, designed to reduce the $m$-sharpness.

This paper studies generalization gap from a different perspective. Noting that the standard procedure for neural network training is *stochastic* so that a different instance leads to a different model, we first present a theoretical analysis in which the bound of generalization gap depends on *inconsistency* and *instability of model outputs*, conceptually described as follows. Let $P$ be a stochastic training procedure, e.g., minimization of the cross-entropy loss by stochastic gradient descent (SGD) starting from random initialization with a certain combination of hyperparameters. It can be regarded as a random function that maps a set of labeled training data as its input to model parameter as its output. We say procedure $P$ has high *inconsistency* of model outputs if the predictions made by the models trained with $P$ using the *same training data* are very different from one another in expectation over the underlying (but unknown) data distribution. We also say $P$ has high *instability* of model outputs if *different* sampling of *training data* changes the expected predictions a lot over the underlying data distribution. Although both quantities are discrepancies of model outputs, the sources of the

---

[*]This work was done when the second author was jointly with Google Research.

discrepancies are different. The term *stability* comes from the related concept in the literature [4, 31]. Inconsistency is related to the *disagreement* metric studied in [27, 17, 22], but there are crucial differences between them, as shown later. Both inconsistency and instability of model outputs can be estimated on *unlabeled data*.

Our high-level goal is to find the essential property of models that generalize well; such an insight would be useful for improving training algorithms. With this goal, we empirically studied the connection of inconsistency and instability with the generalization gap. As our bound also depends on a property of model parameter distributions (for which we have theoretical insight but which we cannot easily estimate), we first experimented to find the condition under which inconsistency and instability of model outputs are highly correlated to generalization gap. The found condition – low randomness in the final state – is consistent with the theory, and it covers practically useful settings. We also found that when this condition is met, inconsistency alone is almost as predictive as inconsistency and instability combined. This is a practical advantage since estimation of instability requires multiple training sets and estimation of inconsistency does not. We thus focused on inconsistency, which enabled the use of full-size training data, and studied inconsistency in comparison with sharpness in the context of algorithmic reduction of these two quantities. We observed that while both sharpness reduction and inconsistency reduction lead to better generalization, there are a number of cases where inconsistency and generalization gap are reduced and yet sharpness remains relatively high. We view it as an indication that inconsistency has a stronger (and perhaps more essential) connection with generalization gap than sharpness.

Our contributions are as follows.

- We develop a theory that relates generalization gap to instability and inconsistency of model outputs, which can be estimated on unlabeled data.
- Empirically, we show that instability and inconsistency are strongly predictive of generalization gap in various settings.
- We show that algorithmic encouragement of consistency reduces inconsistency and improves performance, which can lead to further improvement of the state of the art performance.
- Our results provide a theoretical basis for existing methods such as co-distillation and ensemble.

## 2 Theory

The theorem below quantifies generalization gap by three quantities: inconsistency of model outputs, instability of model outputs, and information-theoretic instability of model parameter distributions.

**Notation** Let $f(\theta, x)$ be the output of model $\theta$ on data point $x$ in the form of probability estimates (e.g., obtained by applying softmax). We use the upright bold font for probability distributions. Let $Z = (X, Y)$ be a random variable representing a labeled data point (data point $X$ and label $Y$) with a given unknown distribution $\mathbf{Z}$. Let $Z_n = \{(X_i, Y_i) : i = 1, \ldots, n\}$ be a random variable representing iid training data of size $n$ drawn from $\mathbf{Z}$. Let $\boldsymbol{\Theta}_{P|Z_n}$ be the distribution of model parameters resulting from applying a (typically stochastic) training procedure $P$ to training set $Z_n$.

**Inconsistency, instability, and information-theoretic instability** Inconsistency of model outputs $\mathcal{C}_P$ ('$\mathcal{C}$' for consistency) of training procedure $P$ represents *the discrepancy of outputs among the models trained on the same training data*:

$$\mathcal{C}_P = \mathbb{E}_{Z_n} \mathbb{E}_{\Theta, \Theta' \sim \boldsymbol{\Theta}_{P|Z_n}} \mathbb{E}_X \mathrm{KL}(f(\Theta, X) || f(\Theta', X)) \qquad \text{(Inconsistency of model outputs)}$$

The source of inconsistency could be the random initialization, sampling of mini-batches, randomized data augmentation, and so forth.

To define instability of model outputs, let $\bar{f}_{P|Z_n}(x)$ be the expected outputs (for data point $x$) of the models trained by procedure $P$ on training data $Z_n$: $\bar{f}_{P|Z_n}(x) = \mathbb{E}_{\Theta \sim \boldsymbol{\Theta}_{P|Z_n}} f(\Theta, x)$. Instability of model outputs $\mathcal{S}_P$ ('$\mathcal{S}$' for stability) of procedure $P$ represents *how much the expected prediction $\bar{f}_{P|Z_n}(x)$ would change with change of training data*:

$$\mathcal{S}_P = \mathbb{E}_{Z_n, Z'_n} \mathbb{E}_X \mathrm{KL}(\bar{f}_{P|Z_n}(X) || \bar{f}_{P|Z'_n}(X)) \qquad \text{(Instability of model outputs)}$$

Finally, the instability of model parameter distributions $\mathcal{I}_P$ is the mutual information between the random variable $\Theta_P$, which represents the model parameters produced by procedure $P$, and the random variable $Z_n$, which represents training data. $\mathcal{I}_P$ quantifies the dependency of the model parameters on the training data, which is also called *information theoretic (in)stability* in [38]:

$$\mathcal{I}_P = I(\Theta_P; Z_n) = \mathbb{E}_{Z_n} \mathrm{KL}(\mathbf{\Theta}_{P|Z_n} || \mathbb{E}_{Z_n'} \mathbf{\Theta}_{P|Z_n'}) \quad \text{(Instability of model parameter distributions)}$$

The second equality follows from the well-known relation of the mutual information to the KL divergence. The rightmost expression might be more intuitive, which essentially represents *how much the model parameter distribution would change with change of training data*.

Note that inconsistency and instability of model outputs can be estimated on *unlabeled data*. $\mathcal{I}_P$ cannot be easily estimated as it involves distributions over the entire model parameter space; however, we have theoretical insight from its definition. (More precisely, it is possible to estimate $\mathcal{I}_P$ by sampling in theory, but practically, it is not quite possible to make a reasonably good estimate of $\mathcal{I}_P$ for deep neural networks with a reasonably small amount of computation.)

**Theorem 2.1.** *Using the definitions above, we consider a Lipschitz loss function $\phi(f, y) \in [0, 1]$ that satisfies $|\phi(f, y) - \phi(f', y)| \leq \frac{\gamma}{2} \|f - f'\|_1$, where $f$ and $f'$ are probability estimates and $\gamma/2 > 0$ is the Lipschitz constant. Let $\psi(\lambda) = \frac{e^\lambda - \lambda - 1}{\lambda^2}$, which is an increasing function. Let $\mathcal{D}_P = \mathcal{C}_P + \mathcal{S}_P$. Let $\Phi_{\mathbf{Z}}(\theta)$ be test loss: $\Phi_{\mathbf{Z}}(\theta) = \mathbb{E}_{Z=(X,Y)} \phi(f(\theta, X), Y)$. Let $\Phi(\theta, Z_n)$ be empirical loss on $Z_n = \{(X_i, Y_i) | i = 1, \cdots, n\}$: $\Phi(\theta, Z_n) = \frac{1}{n} \sum_{i=1}^n \phi(f(\theta, X_i), Y_i)$. Then for a given training procedure $P$, we have*

$$\mathbb{E}_{Z_n} \mathbb{E}_{\Theta \sim \mathbf{\Theta}_{P|Z_n}} \left[ \Phi_{\mathbf{Z}}(\Theta) - \Phi(\Theta, Z_n) \right] \leq \inf_{\lambda > 0} \left[ \gamma^2 \psi(\lambda) \lambda \mathcal{D}_P + \frac{\mathcal{I}_P}{\lambda n} \right].$$

The theorem indicates that the upper bound of generalization gap depends on the three quantities, instability of two types and inconsistency, described above. As a sanity check, note that right after random initialization, generalization gap of the left-hand side is zero, and $\mathcal{I}_P$ and $\mathcal{S}_P$ in the right-hand side are also zero, which makes the right-hand side zero as $\lambda \to 0$. Also note that this analysis is meant for stochastic training procedures. If $P$ is a deterministic function of $Z_n$ and not constant, the mutual information $\mathcal{I}_P$ would become large while $\mathcal{D}_P > 0$, which would make the bound loose.

Intuitively, when training procedure $P$ is more random, model parameter distributions are likely to be flatter (less concentrated) and less dependent on the sampling of training data $Z_n$, and so $\mathcal{I}_P$ is lower. However, high randomness would raise inconsistency of the model outputs $\mathcal{C}_P$, which would raise $\mathcal{D}_P$. Thus, the theorem indicates that there should be a trade-off with respect to the randomness of the training procedure.

The style of this general bound follows from the recent information theoretical generalization analyses of stochastic machine learning algorithms [38, 30, 28] that employ $\mathcal{I}_P$ as a complexity measure, and such results generally hold in expectation. However, unlike the previous studies, we obtained a more refined Bernstein-style generalization result. Moreover, we explicitly incorporate inconsistency and instability of model outputs into our bound and show that smaller inconsistency and smaller instability lead to a smaller generalization bound on the right-hand side. In particular, if $\mathcal{I}_P$ is relatively small so that we have $\mathcal{I}_P \leq n\gamma^2 \mathcal{D}_P$, then setting $\lambda = \sqrt{\mathcal{I}_P/(n\gamma^2 \mathcal{D}_P)}$ and using $\psi(\lambda) < 1$ for $\lambda \leq 1$, we obtain a simpler bound

$$\mathbb{E}_{Z_n} \mathbb{E}_{\Theta \sim \mathbf{\Theta}_{P|Z_n}} \left[ \Phi_{\mathbf{Z}}(\Theta) - \Phi(\Theta, Z_n) \right] \leq 2\gamma \sqrt{\frac{\mathcal{D}_P \mathcal{I}_P}{n}}.$$

**Relation to *disagreement*** It was empirically shown in [27, 17] that with models trained to zero training error, *disagreement* (in terms of classification decision) of identically trained models is approximately equal to test error. When measured for the models that share training data, disagreement is closely related to inconsistency $\mathcal{C}_P$ above, and it can be expressed as $\mathbb{E}_{Z_n} \mathbb{E}_{\Theta, \Theta' \sim \mathbf{\Theta}_{P|Z_n}} \mathbb{E}_X \mathbb{I}[c(\Theta, X) \neq c(\Theta', X)]$, where $c(\theta, x)$ is classification decision $c(\theta, x) = \arg\max_i f(\theta, x)[i]$ and $\mathbb{I}$ is the indicator function $\mathbb{I}[u] = 1$ if $u$ is true and 0 otherwise. In spite of the similarity, in fact, the behavior of disagreement and inconsistency $\mathcal{C}_P$ can be quite different. Disagreement is equivalent to sharpening the prediction to a one-hot vector and then taking 1-norm of the difference: $\mathbb{I}[c(\theta, x) \neq c(\theta', x)] = \frac{1}{2} \|\text{onehot}(f(\theta, x)) - \text{onehot}(f(\theta', x))\|_1$. This ultimate

sharpening makes inconsistency and disagreement very different when the confidence-level of $f(\theta, x)$ is low. This means that disagreement and inconsistency would behave more differently on more complex data (such as ImageNet) on which it is harder to train models to a state of high confidence. In Figure 10 (Appendix) we show an example of how different the behavior of disagreement and inconsistency can be, with ResNet-50 trained on 10% of ImageNet.

## 3 Empirical study

Our empirical study consists of three parts. As the bound depends not only $\mathcal{D}_P (= \mathcal{C}_P + \mathcal{S}_P)$ but also $\mathcal{I}_P$, we first seek the condition under which $\mathcal{D}_P$ is predictive of generalization gap (Section 3.1). Noting that, when the found condition is met, inconsistency $\mathcal{C}_P$ alone is as predictive of generalization gap as $\mathcal{D}_P$, Section 3.2 focuses on inconsistency and shows that inconsistency is predictive of generalization gap in a variety of realistic settings that use full-size training sets. Finally, Section 3.3 reports on the practical benefit of encouraging low inconsistency in algorithmic design. The details for reproduction are provided in Appendix C.

**Remark: Predictiveness of $\mathcal{D}_P$ is relative** Note that we are interested in how well the *change* in $\mathcal{D}_P$ matches the *change* in generalization gap; thus, evaluation of the relation between $\mathcal{D}_P$ and generalization gap always involves *multiple* training procedures to observe the *changes/differences*. Given set $S$ of training procedures, we informally say $\mathcal{D}_P$ *is predictive of generalization gap for $S$ if the relative smallness/largeness of $\mathcal{D}_P$ of the procedures in $S$ essentially coincides with the relative smallness/largeness of their generalization gap*.

### 3.1 On the predictiveness of $\mathcal{D}_P=$ Inconsistency $\mathcal{C}_P+$ Instability $\mathcal{S}_P$

Inconsistency $\mathcal{C}_P$ and instability $\mathcal{S}_P$ were estimated as follows. For each training procedure $P$ (identified by a combination of hyperparameters such as the learning rate and training length), we trained $J$ models on each of $K$ disjoint training sets. That is, $K \times J$ models were trained with each procedure $P$. The expectation values involved in $\mathcal{C}_P$ and $\mathcal{S}_P$ were estimated by taking the average; in particular, the expectation over data distribution $\mathbf{Z}$ was estimated on the held-out unlabeled data disjoint from training data or test data. Disagreement was estimated similarly. $(K, J)$ was set to (4,8) for CIFAR-10/100 and (4,4) for ImageNet, and the size of each training set was set to 4K for CIFAR-10/100 and 120K (10%) for ImageNet.

#### 3.1.1 Results

We start with the experiments that varied the learning rate and the length of training fixing anything else to see whether the change of $\mathcal{D}_P$ is predictive of the change of generalization gap caused by the change of these two hyperparameters. To avoid the complex effects of learning rate decay, we trained models with SGD with a constant learning rate (*constant SGD* in short).

**Constant SGD with iterate averaging (Fig 1,3)** Although constant SGD could perform poorly by itself, constant SGD with *iterate averaging* is known to be competitive [29, 16]. Figure 1 shows $\mathcal{D}_P$ ($x$-axis) and generalization gap ($y$-axis) of the training procedures that performed constant SGD with iterate averaging. Iterate averaging was performed by taking the exponential moving average with momentum 0.999. Each point represents a procedure (identified by the combination of the learning rate and training length), and the procedures with the same learning rate are connected by a line in the increasing order of training length. A *positive correlation* is observed between $\mathcal{D}_P$ and generalization gap on all three datasets. Figure 3 plots generalization gap (left) and $\mathcal{D}_P$ (right) on the $y$-axis and training loss on the $x$-axis for the same procedures as in Figure 1; observe that the up and down of $\mathcal{D}_P$ is remarkably similar to the up and down of generalization gap. Also note that on CIFAR-10/100, sometimes generalization gap first goes up and then starts coming down towards the low-training-loss end of the lines (increase of training loss in some cases is due to the effects of weight decay). This '*turning around*' of generalization gap (from going up to going down) in Figure 3 shows up as the '*curling up*' of the upper-right end of the lines in Figure 1. It is interesting to see that $\mathcal{D}_P$ and generalization gap are matching at such a detailed level.

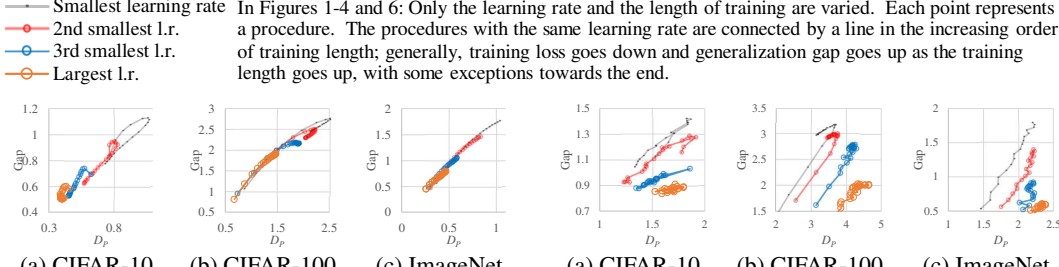

In Figures 1-4 and 6: Only the learning rate and the length of training are varied. Each point represents a procedure. The procedures with the same learning rate are connected by a line in the increasing order of training length; generally, training loss goes down and generalization gap goes up as the training length goes up, with some exceptions towards the end.

(a) CIFAR-10  (b) CIFAR-100  (c) ImageNet

Figure 1: $\mathcal{D}_P$ estimates ($x$-axis) and generalization gap ($y$-axis). SGD with a **constant learning rate** and **iterate averaging**. Only the learning rate and training length were varied. A positive correlation is observed.

(a) CIFAR-10  (b) CIFAR-100  (c) ImageNet

Figure 2: $\mathcal{D}_P$ estimates ($x$-axis) and generalization gap ($y$-axis). **Constant learning rates. No iterate averaging**. $\mathcal{D}_P$ is predictive of generalization gap only for the procedures that share the learning rate.

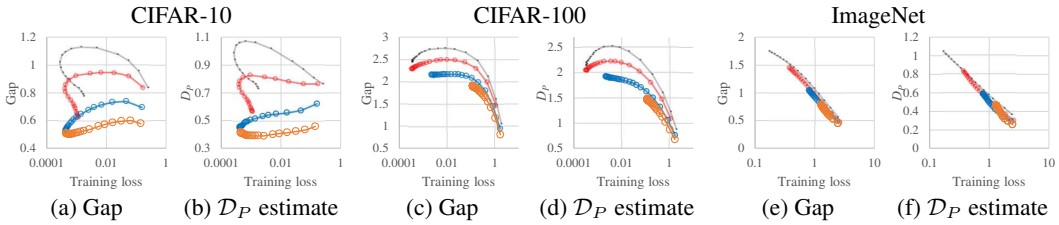

CIFAR-10  CIFAR-100  ImageNet

(a) Gap  (b) $\mathcal{D}_P$ estimate  (c) Gap  (d) $\mathcal{D}_P$ estimate  (e) Gap  (f) $\mathcal{D}_P$ estimate

Figure 3: Up and down of generalization gap ($y$-axis; left) and $\mathcal{D}_P$ ($y$-axis; right) as training proceeds. The $x$-axis is training loss. SGD with a constant learning rate with iterate averaging. The analyzed models and the legend are the same as in Fig 1. For each dataset, the left graph looks very similar to the right graph; up and down of $\mathcal{D}_P$ as training proceeds is a good indicator of up and down of generalization gap.

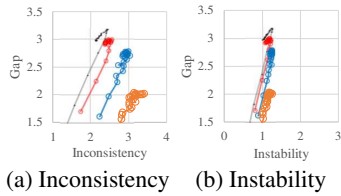

(a) Inconsistency  (b) Instability

Figure 4: Inconsistency and instability ($x$-axis) and generalization gap ($y$-axis). Same procedures and legend as in Fig 2. Instability is relatively unaffected by the learning rate. CIFAR-100. Similar results on CIFAR-10 and ImageNet (Figure 11 in the Appendix).

**Constant SGD without iterate averaging (Fig 2,4)** While $\mathcal{D}_P$ is clearly predictive of generalization gap in the setting above, the results in Figure 2 are more complex. This figure shows $\mathcal{D}_P$ ($x$-axis) and generalization gap ($y$-axis) for the training procedures that performed constant SGD and did *not* perform iterate averaging. As before, only the learning rate and training length were varied. In Figure 2, we observe that $\mathcal{D}_P$ is predictive *only* for those which share the learning rate. For fixed generalization gap, $\mathcal{D}_P$ is *larger* (instead of being the same) for a *larger* learning rate. Inspection of inconsistency and instability reveals that larger learning rates raise inconsistency although instability is mostly unaffected (see Fig 4). This is apparently the effect of high randomness/noisiness/uncertainty of the final state. As the learning rate is constant, SGD updates near the end of training bounce around the local minimum, and the random noise in the gradient of the last mini-batch has a substantial influence on the final model parameter. The influence of this noise is amplified by larger learning rates. On the other hand, $\mathcal{I}_P$ is likely to be lower with higher randomness since higher randomness should make model parameter distributions flatter (less concentrated) and less dependent on the sampling of training data $Z_n$. This means that in this case (i.e., where the final randomness is high and varies a lot across the procedures), the interaction of $\mathcal{I}_P$ and $\mathcal{D}_P$ in the theoretical bound is complex, and $\mathcal{D}_P$ alone could be substantially less predictive than what is indicated by the bound.

**For $\mathcal{D}_P$ to be predictive, final randomness should be equally low (Fig 5 (a)–(c))** Therefore, we hypothesized and empirically confirmed that for $\mathcal{D}_P$ to be predictive, the degrees of final randomness/uncertainty/noisiness should be equally low, which can be achieved with either a vanishing learning rate or iterate averaging. This means that, fortunately, $\mathcal{D}_P$ is mostly predictive for high-performing procedures that matter in practice.

Figure 5 (a)–(c) plot $\mathcal{D}_P$ ($x$-axis) and generalization gap ($y$-axis) for the procedures that satisfy the condition. In particular, those trained on CIFAR-10 include a wide variety of procedures that differ

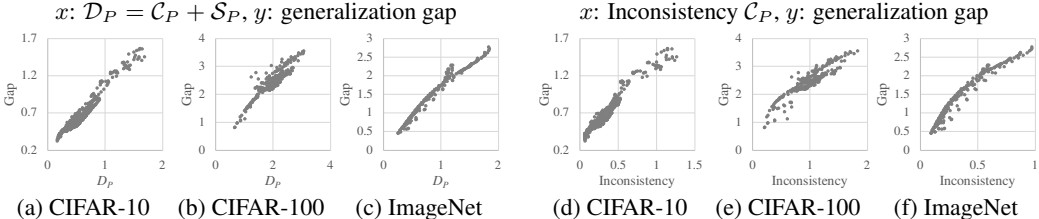

(a) CIFAR-10     (b) CIFAR-100     (c) ImageNet     (d) CIFAR-10     (e) CIFAR-100     (f) ImageNet

Figure 5: (a)–(c) $\mathcal{D}_P$ ($x$-axis) and generalization gap ($y$-axis). (d)–(f) $\mathcal{C}_P$ ($x$-axis) and generalization gap ($y$-axis). Both $\mathcal{D}_P$ and $\mathcal{C}_P$ are predictive of generalization gap for training procedures with iterate averaging or a vanishing learning rate (so that final randomness is low), irrespective of differences in the setting. In particular, the CIFAR-10 results include the training procedures that differ in network architectures, mini-batch sizes, data augmentation, weight decay parameters, learning rate schedules, learning rates and training lengths.

in network architectures, mini-batch sizes, presence/absence of data augmentation, learning rate schedules (constant or vanishing), weight decay parameters, in addition to learning rates and training lengths. A positive correlation between $\mathcal{D}_P$ and generalization gap is observed on all three datasets.

**Inconsistency $\mathcal{C}_P$ vs. $\mathcal{D}_P$ (Fig 5 (d)–(f))** Figure 5 (d)–(f) show that inconsistency $\mathcal{C}_P$ is almost as predictive as $\mathcal{D}_P$ when the condition of low randomness of the final states is satisfied. This is a practical advantage since unlike instability $\mathcal{S}_P$, inconsistency $\mathcal{C}_P$ does not require multiple training sets for estimation.

**Disagreement (Fig 6,7)** For the same procedures/models as in Figures 1 and 5, we show the relationship between disagreement and test error in Figures 6 and 7, respectively. We chose test error instead of generalization gap for the $y$-axis since the previous finding was 'Disagreement $\approx$ Test error'. The straight lines are $y = x$. The relation in Fig 6 (a) and Fig 7 (a) is close to equality, but the relation appears to be more complex in the others. The procedures plotted in Fig 7 are those which satisfy the condition for $\mathcal{D}_P$ to be predictive of generalization gap; thus, the results indicate that the condition for disagreement to be predictive of test error is apparently different.

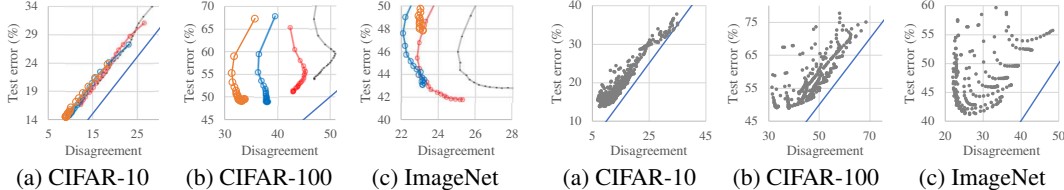

(a) CIFAR-10     (b) CIFAR-100     (c) ImageNet     (a) CIFAR-10     (b) CIFAR-100     (c) ImageNet

Figure 6: Disagreement ($x$-axis) and test error ($y$-axis). SGD with a constant learning rate and iterate averaging. Same models and legend as in Fig 1. "Disagreement$\approx$Test error" of the previous studies does not quite hold in (b) and (c).

Figure 7: Disagreement ($x$-axis) and test error ($y$-axis). The models are the same as in Fig 5, which satisfy the condition for $\mathcal{D}_P$ to be predictive of generalization gap. The results indicate that the condition for disagreement to be predictive of test error is different.

Note that while the previous empirical study of disagreement [27, 17] focused on the models with zero training error, we studied a wider range of models. Unlike CIFAR-10/100 (on which zero training error can be quickly achieved), a common practice for datasets like ImageNet is *budgeted training* [25] instead of training to zero error. Also note that zero training error does not necessarily lead to the best performance, which, for example, has been observed in the ImageNet experiments of this section.

## 3.2 On the predictiveness of inconsistency $\mathcal{C}_P$: from an algorithmic perspective

Based on the results above, this section focuses on inconsistency $\mathcal{C}_P$, which enables experiments with larger training data, and studies the behavior of inconsistency in comparison with sharpness from an algorithmic perspective.

Table 1:  Datasets. Mostly full-size natural images and 2 texts.

| Name | #class | #train | | #dev | #test |
|---|---|---|---|---|---|
| | | provided | used | | |
| ImageNet [7] | 1000 | 1.28M | 1.27M | 10K | 50K |
| Food101 [3] | 101 | 75750 | 70700 | 5050 | 25250 |
| Dogs [21] | 120 | 12000 | 10800 | 1200 | 8580 |
| Cars [23] | 196 | 8144 | 7144 | 1000 | 8041 |
| CIFAR-10 [24] | 10 | 50000 | 4000 | 5000 | 10000 |
| MNLI [36] | 3 | 392702 | 382902 | 9800 | 9815 |
| QNLI [36] | 2 | 104743 | 99243 | 5500 | 5463 |

Table 2:  Networks.

| Network | #param | Case |
|---|---|---|
| ResNet-50 [12] | 24M | 1 |
| ViT-S/32 [9] | 23M | 2 |
| ViT-B/16 [9] | 86M | 3 |
| Mixer-B/16 [35] | 59M | 4 |
| WRN-28-2 [40] | 1.5M | 5 |
| EN-B0 [33] | 4.2M | 6,7 |
| roberta-base[26] | 124.6M | 8,9 |
| ResNet-18 [12] | 11M | 10 |

**Training objectives**   Parallel to the fact that SAM seeks flatness of training loss landscape, a meta-algorithm *co-distillation* (named by [1], and closely related to deep mutual learning [42], summarized in Algorithm 1 in the Appendix) encourages consistency of model outputs. It simultaneously trains two (or more) models with two (or more) different random sequences while penalizing the inconsistency between the predictions of the two models. This is typically described as 'teaching each other', but we take a different view of consistency encouragement. Note that the penalty term merely '*encourages*' low inconsistency. Since inconsistency by definition depends on the unknown data distribution, it cannot be directly minimized by training. The situation is similar to minimizing loss on the training data with the hope that loss will be small on unseen data. We analyzed the models that were trained with one of the following training objectives (shown with the abbreviations used below):

- *'Standard'*: the standard cross-entropy loss.
- *'Consist.'*: the standard loss with encouragement of consistency.
- *'Flat.'*: the SAM objective, which encourages flatness.
- *'Consist+Flat'*: encouraging both consistency and flatness. Coupling two instances of SAM training with the inconsistency penalty term.

**Cases**   We experimented with ten combinations (*cases*) of dataset, network architecture, and training scenario. Within each case, we fixed the basic settings (e.g., weight decay, learning rate) to the ones known to perform well from the previous studies, and only varied the training objectives so that for each case we had at least four stochastic training procedures distinguished by the four training objectives (for some cases, we had more than four due to testing multiple values of $\rho$ for SAM). We obtained four models (trained with four different random sequences) per training procedure. Tables 1 and 2 show the datasets and network architectures we used, respectively. Note that as a result of adopting high-performing settings, all the cases satisfy the condition of low final randomness.

### 3.2.1   Results

As in the previous section, we quantify the generalization gap by the difference of test loss from training loss (note that the loss is the cross-entropy loss; the inconsistency penalty or anything else is not included); the inconsistency values were again measured on the held-out unseen *unlabeled* data, disjoint from both training data and test data. Two types of sharpness values were measured, 1-sharpness (per-example loss difference in the adversarial direction) of [10] and the magnitude of the loss Hessian (measured by the largest eigenvalue), which represent the sharpness of training loss landscape and have been shown to correlate to generalization performance [10].

**Inconsistency correlates to generalization gap (Figure 8)**   Figure 8 shows inconsistency or sharpness values ($x$-axis) and the generalization gap ($y$-axis). Each point in the figures represents a model; i.e., in this section, we show model-wise quantities (instead of procedure-wise), simulating the model selection setting. (Model-wise inconsistency for model $\theta$ trained on $Z_n$ with procedure $P$ is $\mathbb{E}_{\Theta \sim \Theta_{P|Z_n}} \mathbb{E}_X \mathrm{KL}(f(\Theta, X) || f(\theta, X))$.) All quantities are standardized so that the mean is zero and the standard deviation is 1. In the figures, we distinguish the four training objectives. In Figure 8 (b) and (c), we confirm, by comparing the 'Flat.' models ($\times$) with the baseline standard models ($+$), that encouragement of flatness (by SAM) indeed reduces sharpness ($x$-axis) as well as the generalization gap ($y$-axis). This serves as a sanity check of our setup.

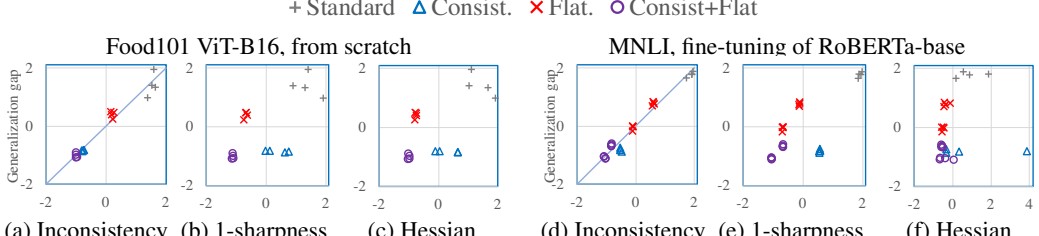

**Figure 8:** Inconsistency and sharpness ($x$-axis) and generalization gap ($y$-axis). Each point represents a model. Note that each graph plots 16–20 points, and some of them are overlapping.

We observe in Figure 8 that inconsistency shows a clear positive correlation with the generalization gap, but the correlation of sharpness with generalization gap is less clear (e.g., in (b)–(c), the generalization gap of the 'Consist.' models ($\triangle$) is much smaller than that of the 'Flat.' models ($\times$), but their sharpness is larger). This is a general trend observed across multiple network architectures (ResNet, Transformers, MLP-Mixer, and so on), multiple datasets (images and texts), and multiple training scenarios (from scratch, fine-tuning, and distillation); the Appendix provides the figures for all 10 cases. Moreover, we found that while both sharpness reduction and inconsistency reduction lead to better generalization, in a number of cases inconsistency and generalization gap are reduced and yet sharpness remains relatively high. We view this phenomenon as an indication that *inconsistency has a stronger (and perhaps more essential) connection with generalization gap than sharpness*.

**Inconsistency is more predictive of generalization gap than sharpness (Table 3)**  Motivated by the linearity observed in the figure above, we define a linear predictor of the generalization gap that takes the inconsistency or sharpness value as input. We measure the predictive power of the metrics through the prediction performance of this linear predictor trained with least square minimization. To make the inconsistency and sharpness values comparable, we standardize them and also generalization gap. For each case, we evaluated the metrics by performing the leave-one-out cross validation on the given set of models (i.e., perform least squares using all models but one and evaluate the obtained linear predictor on the left-out model; do this $k$ times for $k$ models and take the average). The results in Table 3 show that inconsistency outperforms sharpness, and the superiority still generally holds even when sharpness is given an 'unfair' advantage of additional *labeled* data.

**Table 3:** Generalization gap prediction error. The leave-one-out cross validation results, which average the generalization gap prediction residuals, are shown. See Table 2 for the networks used for each case. Smaller is better, and a value near 1.0 is very poor. Inconsistency: estimated on unlabeled data, 1-sharpness and Hessian: sharpness of training loss landscape. Inconsistency outperforms the sharpness metrics. Noting that inconsistency had access to additional data, even though *unlabeled*, we also show in the last 2 rows sharpness of test loss landscape estimated on the development data (held-out *labeled* data), which gives the sharpness metrics an 'unfair' advantage by providing them with additional *labels*. With this advantage, the predictiveness of sharpness mostly improves; however, still, inconsistency is generally more predictive. The **bold** font indicates that the difference is statistically significant at 95% confidence level against all others including the last 2 rows. The *italic* font indicates the best but not statistically significant.

| Case# | 1 | 2 | 3 | 4 | 5 | 6 | 7 | 8 | 9 | 10 |
|---|---|---|---|---|---|---|---|---|---|---|
| Dataset | ImageNet | | Food101 | C10 | Dog | Car | Mnli | Qnli | Food |
| Training scenario | From scratch | | | | | Fine-tuning | | | | Distill. |
| Inconsistency | *0.13* | **0.10** | **0.17** | **0.27** | 0.15 | **0.09** | 0.19 | **0.15** | **0.18** | **0.19** |
| 1-sharpness | 0.21 | 0.48 | 0.75 | 0.74 | 0.84 | 0.34 | 0.24 | 0.58 | 0.98 | 0.75 |
| Hessian | 0.50 | 1.00 | 0.77 | 0.72 | 0.78 | 0.70 | 0.68 | 0.59 | 0.87 | 0.59 |
| Giving the sharpness metrics an 'unfair' advantage by providing additional *labels*: | | | | | | | | | | |
| 1-sharpness of dev. loss | 0.14 | 0.64 | 0.47 | 0.40 | 0.72 | 0.22 | *0.14* | 0.36 | 0.37 | 0.62 |
| Hessian of dev. loss | 0.45 | 0.82 | 0.58 | 0.75 | *0.13* | 0.27 | 0.15 | 0.91 | 0.81 | 0.89 |

## 3.3   Practical impact: algorithmic consequences

The results above suggest a strong connection of inconsistency of model outputs with generalization, which also suggests the importance of seeking consistency in the algorithmic design. To confirm this

point, we first experimented with two methods (ensemble and distillation) and found that in both, encouraging low inconsistency in every stage led to the best test error. Note that in this section we report test error instead of generalization gap in order to show the practical impact.

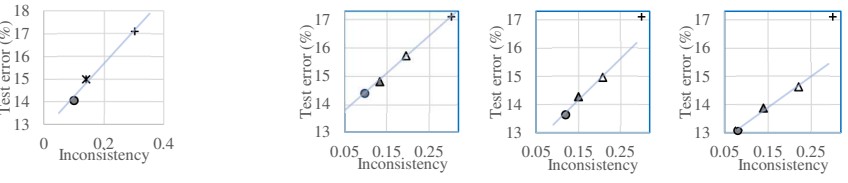

(a) Ensemble      (b) Distillation. Teacher's knowledge level increases from left to right.

Figure 9: Inconsistency and test error with ensemble and distillation.
(a) Ensemble reduces inconsistency ($x$-axis) and test error ($y$-axis). $+$ Non-ensemble (baseline), $*$ Ensemble of standard models, $\bullet$ Ensemble of the models trained with consistency encouragement. RN18 on Food101.
(b) Test error ($y$-axis) and inconsistency of distilled models ($x$-axis). Without unlabeled data. $+$ No distillation, $\triangle$ Standard distillation, $\blacktriangle$ Consistency encouragement for the student, $\bullet$ Consistency encouragement for both the student and teacher. The best performance is obtained when consistency is encouraged for both the teacher and student. Food101. Student: RN18, Teacher: RN18 (left), RN50 (middle), and fine-tuning of ImageNet-trained ENB0 (right). In both (a) and (b), the average of 4 models is shown; see Appendix for the standard deviations.

**Ensemble (Fig 9 (a))** Figure 9 (a) shows that, compared with non-ensemble models ($+$), ensemble models (averaging logits of two models) reduce both inconsistency ($x$-axis) and test error ($y$-axis). It is intuitive that averaging the logits would cancel out the dependency on (or particularities of) the individual random sequences inherent in each single model; thus, outputs of ensemble models are expected to become similar and so *consistent* (as empirically observed). Moreover, we presume that reduction of inconsistency is the mechanism that enables ensemble to achieve better generalization. The best test error was achieved when the ensemble was made by two models trained with consistency encouragement ($\bullet$ in Fig 9 (a)).

**Distillation (Fig 9 (b))** We experimented with distillation [13] with encouragement of consistency for the teacher model and/or the student model. Figure 9 (b) shows the test error ($y$-axis) and the inconsistency ($x$-axis). The standard model ($+$) serves as the baseline. The knowledge level of teachers increases from left to right, and within each graph, when/whether to encourage consistency differs from the point to point. The main finding here is that at all the knowledge levels of the teacher, both inconsistency and test error go down as the pursuit of consistency becomes more extensive; in each graph, the points for the *distilled* models form a nearly straight line. (The line appears to shift away from the baseline ($+$) as the amount of external knowledge of the teacher increases. We conjecture this could be due to the change in $\mathcal{I}_P$.) While distillation alone reduces inconsistency and test error, the best results are obtained by encouraging consistency at every stage of training (i.e., training both the teacher and student with consistency encouragement). The results underscore the importance of inconsistency reduction for better performance.

**On the pursuit of the state of the art (Table 4,5)** We show two examples of obtaining further improvement of state-of-the-art results by adding consistency encouragement. The first example is semi-supervised learning with CIFAR-10. Table 4 shows that the performance of a state-of-the-art semi-supervised method can be further improved by encouraging consistency between two training instances of this method on the unlabeled data (Algorithm 2 in the Appendix). The second example is transfer learning. We fine-tuned a public ImageNet-trained EfficientNet with a more focused dataset Food101. Table 5 shows that encouraging consistency between two instances of SAM training improved test error. These examples demonstrate the importance of consistency encouragement in the pursuit of the state of the art performance.

## 4 Limitations and discussion

Theorem 2.1 assumes a bounded loss $\phi(f, y) \in [0, 1]$ though the standard loss for classification is the cross-entropy loss, which is unbounded. This assumption can be removed by extending the theorem to a more complex analysis with other moderate assumptions. We, however, chose to present the simpler and so more intuitive analysis with a bounded loss. As noted above, this theorem is intended

Table 4: CIFAR-10 #train=4K, #unlabeled=41K, WRN28-2. Average and standard deviation of 5 runs.

| | Methods | Error (%) |
|---|---|---|
| Copied from [32] | MixMatch [2] | 6.42 |
| | UDA [37] | 4.88 |
| | FixMatch [32] | 4.26 |
| Our results | FixMatch | $4.25_{\pm 0.15}$ |
| | UDA (no sharpening) | $4.33_{\pm 0.10}$ |
| | UDA + 'Consist.' | $\mathbf{3.95}_{\pm 0.12}$ |

RandAugment [6] was used for UDA and FixMatch.

Table 5: Food101, EfficientNet-B4. The average and standard deviation of 6 models are shown.

| | Methods | Error (%) |
|---|---|---|
| Previous results | EN-B4 [33] | 8.5 |
| | EN-B7 [10] | 7.17 |
| | EN-B7 SAM [10] | 7.02 |
| Our results | EN-B4 SAM † | $6.00_{\pm 0.05}$ |
| | EN-B4 SAM + 'Consist.' | $\mathbf{5.77}_{\pm 0.04}$ |

† The improvement over EN-B7 SAM of [10] is due to the difference in the basic setting; see Appendix.

for stochastic training procedures. The bound may not be useful for deterministic procedures, and this characteristics is not unique to our analysis but shared by the previous information-theoretic analysis of stochastic machine learning algorithms [38, 30, 28].

We acknowledge that due to resource constraints, there was a limitation to the diversity of the training procedures we empirically analyzed. In particular, our study of inconsistency and instability in Section 3.1 started with simpler cases of constant learning rate SGD and later included the cases of learning rate decay (Figure 5); consequently, the types of the training procedures studied in Section 3.1 were skewed towards the constant learning rate SGD. The models analyzed in Figure 5 were less diverse with CIFAR-100 and ImageNet than with CIFAR-10, as noted above, and again this asymmetry was due to the resource constraints. On the other hand, the models analyzed in this work are in a sense more diverse than the previous empirical studies [27, 17] of the disagreement metric (related to our inconsistency), which were restricted to the models with near zero train error.

In Section 3.2, we studied inconsistency from the algorithmic perspective in comparison with the sharpness metrics. We chose sharpness for comparison due to its known correlation to generalization performance and the existence of the algorithm (SAM) to reduce it. We acknowledge that there can be other metrics that are predictive of generalization gap.

Although the correlation of the disagreement metric with generalization gap is relatively poor in the settings of Section 3.1 (e.g., Figure 10), it is plausible that disagreement can be as predictive of generalization gap as inconsistency in some settings, for example, when the confidence level of predictions is sufficiently high since in that case, Theorem 2.1 should provide a theoretical basis also for disagreement though indirectly (details are provided in Appendix D.3). Moreover, for improving stochastic training of deep neural networks, it would be useful to understand the connection between generalization performance and discrepancies of model outputs in general (whether instability, inconsistency, disagreement, or something else), and we hope that this work contributes to pushing forward in this direction.

# 5 Conclusion

We presented a theory that relates generalization gap to instability and inconsistency of model outputs, which can be estimated on unlabeled data, and empirically showed that they are strongly predictive of generalization gap in various settings. In particular, inconsistency was shown to be a more reliable indicator of generalization gap than commonly used local flatness. We showed that algorithmic encouragement of consistency reduces inconsistency as well as test error, which can lead to further improvement of the state of art performance. Finally, our results provide a theoretical basis for existing methods such as co-distillation and ensemble.

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

# Appendix

## A Proof of Theorem 2.1

As in the main paper, let $Z = (X, Y)$ be a random variable representing a labeled data point (data point $X$ and label $Y$) with distribution $\mathbf{Z}$. Let $Z_n = \{(X_i, Y_i) : i = 1, 2, \ldots, n\}$ be a random variable representing iid training data of size $n$ drawn from $\mathbf{Z}$.

We have the following lemma.

**Lemma A.1.** *Given an arbitrary model parameter distribution $\mathbf{\Theta}^0$, let*

$$\bar{f}_P(x) = \mathbb{E}_{Z_n'} \bar{f}_{P|Z_n'}(x) = \mathbb{E}_{Z_n'} \mathbb{E}_{\Theta \sim \mathbf{\Theta}_{P|Z_n'}} f(\Theta, x),$$

$$\ell_\theta(z) = \phi(f(\theta, x), y) - \phi(\bar{f}_P(x), y) \text{ where } z = (x, y)$$

*then*

$$-n\mathbb{E}_{Z_n} \mathbb{E}_{\Theta \sim \mathbf{\Theta}_{P|Z_n}} \ln \mathbb{E}_Z \exp(-\lambda \ell_\Theta(Z)) \leq \mathbb{E}_{Z_n} \mathbb{E}_{\Theta \sim \mathbf{\Theta}_{P|Z_n}} \sum_{i=1}^n \lambda \ell_\Theta(X_i, Y_i) + \mathbb{E}_{Z_n} \mathrm{KL}(\mathbf{\Theta}_{P|Z_n} || \mathbf{\Theta}^0).$$

*Proof.* Let $\mathbf{\Theta}_*$ be a model parameter distribution such that

$$\mathbf{\Theta}_* \propto \mathbf{\Theta}^0 \exp\left[\sum_{i=1}^n \left(-\lambda \ell_\Theta(X_i, Y_i) - \ln \mathbb{E}_Z \exp(-\lambda \ell_\Theta(Z))\right)\right].$$

We have

$$\mathbb{E}_{Z_n} \exp\left[\mathbb{E}_{\Theta \sim \mathbf{\Theta}_{P|Z_n}} \sum_{i=1}^n \left(-\lambda \ell_\Theta(X_i, Y_i) - \ln \mathbb{E}_Z \exp(-\lambda \ell_\Theta(Z))\right) - \mathrm{KL}(\mathbf{\Theta}_{P|Z_n} || \mathbf{\Theta}^0)\right]$$

$$\leq \mathbb{E}_{Z_n} \sup_{\mathbf{\Theta}} \exp\left[\mathbb{E}_{\Theta \sim \mathbf{\Theta}} \sum_{i=1}^n \left(-\lambda \ell_\Theta(X_i, Y_i) - \ln \mathbb{E}_Z \exp(-\lambda \ell_\Theta(Z))\right) - \mathrm{KL}(\mathbf{\Theta} || \mathbf{\Theta}^0)\right]$$

$$= \mathbb{E}_{Z_n} \sup_{\mathbf{\Theta}} \mathbb{E}_{\Theta \sim \mathbf{\Theta}^0} \exp\left[\sum_{i=1}^n \left(-\lambda \ell_\Theta(X_i, Y_i) - \ln \mathbb{E}_Z \exp(-\lambda \ell_\Theta(Z))\right) - \mathrm{KL}(\mathbf{\Theta} || \mathbf{\Theta}_*)\right]$$

$$= \mathbb{E}_{Z_n} \mathbb{E}_{\Theta \sim \mathbf{\Theta}^0} \exp\left[\sum_{i=1}^n \left(-\lambda \ell_\Theta(X_i, Y_i) - \ln \mathbb{E}_Z \exp(-\lambda \ell_\Theta(Z))\right)\right] = 1.$$

The first inequality takes $\sup$ over all probability distributions of model parameters. The first equality can be verified using the definition of the KL divergence. The second equality follows from the fact that the supreme is attained by $\mathbf{\Theta} = \mathbf{\Theta}_*$. The last equality uses the fact that $(X_i, Y_i)$ for $i = 1, \ldots, n$ are iid samples drawn from $\mathbf{Z}$. The desired bound follows from Jensen's inequality and the convexity of $\exp(\cdot)$. $\qquad\square$

*Proof of Theorem 2.1.* Using the notation of Lemma A.1, we have

$$\mathbb{E}_{\Theta \sim \mathbf{\Theta}_{P|Z_n}} \ln \mathbb{E}_Z \exp(-\lambda \ell_\Theta(Z)) \leq \mathbb{E}_{\Theta \sim \mathbf{\Theta}_{P|Z_n}} \mathbb{E}_Z \left[\exp(-\lambda \ell_\Theta(Z) - 1\right]$$

$$\leq -\lambda \mathbb{E}_{\Theta \sim \mathbf{\Theta}_{P|Z_n}} \mathbb{E}_Z \ell_\Theta(Z) + \psi(\lambda) \lambda^2 \mathbb{E}_{\Theta \sim \mathbf{\Theta}_{P|Z_n}} \mathbb{E}_Z \ell_\Theta(Z)^2$$

$$\leq -\lambda \mathbb{E}_{\Theta \sim \mathbf{\Theta}_{P|Z_n}} \mathbb{E}_Z \ell_\Theta(Z) + \frac{\gamma^2}{4} \psi(\lambda) \lambda^2 \mathbb{E}_{\Theta \sim \mathbf{\Theta}_{P|Z_n}} \mathbb{E}_X \|f(\Theta, X) - \bar{f}_P(X)\|_1^2. \tag{1}$$

The first inequality uses $\ln u \leq u - 1$. The second inequality uses the fact that $\psi(\lambda)$ is increasing in $\lambda$ and $-\lambda \ell_\theta(z) \leq \lambda$. The third inequality uses the Lipschitz assumption of the loss function.

We also have the following from the triangle inequality of norms, Jensen's inequality, the relationship between the 1-norm and the total variation distance of distributions, and Pinsker's inequality.

$$\mathbb{E}_{Z_n}\mathbb{E}_{\Theta\sim\boldsymbol{\Theta}_{P|Z_n}}\mathbb{E}_X\|f(\Theta,X)-\bar{f}_P(X)\|_1^2$$

$$\leq 2\mathbb{E}_{Z_n}\mathbb{E}_{\Theta\sim\boldsymbol{\Theta}_{P|Z_n}}\mathbb{E}_X\left[\|f(\Theta,X)-\bar{f}_{P|Z_n}(X)\|_1^2+\|\bar{f}_{P|Z_n}(X)-\bar{f}_P(X)\|_1^2\right]$$

$$\leq 2\mathbb{E}_{Z_n}\mathbb{E}_{\Theta,\Theta'\sim\boldsymbol{\Theta}_{P|Z_n}}\mathbb{E}_X\|f(\Theta,X)-f(\Theta',X)\|_1^2+2\mathbb{E}_{Z_n}\mathbb{E}_X\|\bar{f}_{P|Z_n}(X)-\bar{f}_P(X)\|_1^2$$

$$\leq 4\mathbb{E}_{Z_n}\mathbb{E}_{\Theta,\Theta'\sim\boldsymbol{\Theta}_{P|Z_n}}\mathbb{E}_X\text{KL}(f(\Theta,X)\|f(\Theta',X))+2\mathbb{E}_{Z_n}\mathbb{E}_X\|\bar{f}_{P|Z_n}(X)-\bar{f}_P(X)\|_1^2$$

$$=4\mathcal{C}_P+2\mathbb{E}_{Z_n}\mathbb{E}_X\|\bar{f}_{P|Z_n}(X)-\bar{f}_P(X)\|_1^2\leq 4\mathcal{C}_P+2\mathbb{E}_{Z_n}\mathbb{E}_{Z_n'}\mathbb{E}_X\|\bar{f}_{P|Z_n}(X)-\bar{f}_{P|Z_n'}(X)\|_1^2$$

$$\leq 4\mathcal{C}_P+4\mathbb{E}_{Z_n}\mathbb{E}_{Z_n'}\mathbb{E}_X\text{KL}(\bar{f}_{P|Z_n}(X)\|\bar{f}_{P|Z_n'}(X))=4\left(\mathcal{C}_P+\mathcal{S}_P\right)=4\mathcal{D}_P. \tag{2}$$

Using (1), (2), and Lemma A.1, we obtain

$$\lambda\mathbb{E}_{Z_n}\,\mathbb{E}_{\Theta\sim\boldsymbol{\Theta}_{P|Z_n}}\mathbb{E}_Z\left[n\ell_\Theta(Z)-\sum_{i=1}^n\ell_\Theta(X_i,Y_i)\right]\leq n\gamma^2\psi(\lambda)\lambda^2\,\mathcal{D}_P+\mathbb{E}_{Z_n}\,\text{KL}(\boldsymbol{\Theta}_{P|Z_n}\|\boldsymbol{\Theta}^0). \tag{3}$$

Set $\boldsymbol{\Theta}^0=\mathbb{E}_{Z_n'}\boldsymbol{\Theta}_{P|Z_n'}$ so that we have

$$\mathcal{I}_P=\mathbb{E}_{Z_n}\,\text{KL}(\boldsymbol{\Theta}_{P|Z_n}\|\boldsymbol{\Theta}^0). \tag{4}$$

Also note that $\bar{f}_P$ in $\ell_\Theta$ cancels out since $Z_n$ is iid samples of $\mathbf{Z}$, and so using the notation for the test loss and empirical loss defined in Theorem 2.1, we have

$$\lambda\mathbb{E}_{Z_n}\,\mathbb{E}_{\Theta\sim\boldsymbol{\Theta}_{P|Z_n}}\mathbb{E}_Z\left[n\ell_\Theta(Z)-\sum_{i=1}^n\ell_\Theta(X_i,Y_i)\right]=n\lambda\mathbb{E}_{Z_n}\,\mathbb{E}_{\Theta\sim\boldsymbol{\Theta}_{P|Z_n}}\left[\Phi_{\mathbf{Z}}(\Theta)-\Phi(\Theta,Z_n)\right]. \tag{5}$$

(3), (4) and (5) imply the result. $\square$

# B   Additional figures

Figure 10 supplements 'Relation to *disagreement*' at the end of Section 2. It shows an example where the behavior of inconsistency is different from disagreement. Training was done on 10% of ImageNet with the *seed procedure* (tuned to perform well) with training length variations described in Table 6 below. Essentially, in this example, inconsistency goes up like generalization gap, and disagreement goes down like test error and goes up in the end, as training becomes longer.

Figure 11 supplements Figure 4 in Section 3.1. It shows inconsistency and instability of model outputs of the models trained with SGD with constant learning rates without iterate averaging. In this setting of high final randomness, larger learning rates make inconsistency larger while instability is mostly unaffected. By contrast, Figure 12 shows that when final randomness is low (due to iterate averaging in this case), both inconsistency and instability are predictive of generalization gap both within and across the learning rates.

Figure 13–15 supplement Figure 8 in Section 3.2. These figures show the relation of inconsistency and sharpness to generalization gap. Note that each graph has at least 16 points, and some of them (typically for the models trained with the same procedure) are overlapping. Inconsistency shows a stronger correlation with generalization gap than sharpness does.

# C   Experimental details

All the experiments were done using GPUs (A100 or older).

## C.1   Details of the experiments in Section 3.1

The goal of the experiments reported in Section 3.1 was to find whether/how the predictiveness of $\mathcal{D}_P$ is affected by the diversity of the training procedures in comparison. To achieve this goal, we chose the training procedures to experiment with in the following four steps.

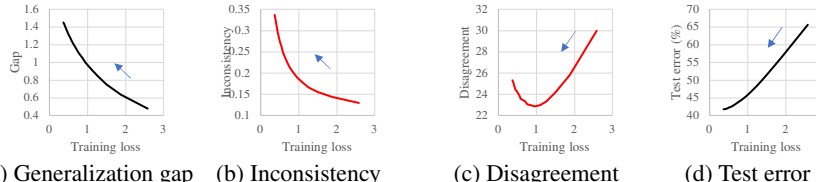

(a) Generalization gap   (b) Inconsistency   (c) Disagreement   (d) Test error

Figure 10: Inconsistency $\mathcal{C}_P$ and disagreement ($y$-axis) in comparison with generalization gap and test error ($y$-axis). The $x$-axis is train loss. The arrows indicate the direction of training becoming longer. Each point is the average of 16 instances. Training was done on 10% of ImageNet with the *seed procedure* (tuned to perform well) with training length variations; see Table 6. In this example, essentially, inconsistency goes up like generalization gap, and disagreement goes down like test error and goes up in the end, as training becomes longer.

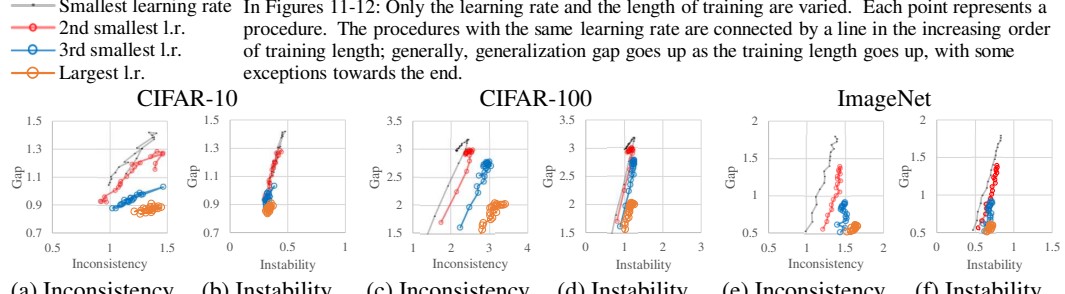

(a) Inconsistency   (b) Instability   (c) Inconsistency   (d) Instability   (e) Inconsistency   (f) Instability

Figure 11: Inconsistency $\mathcal{C}_P$ (left) and instability $\mathcal{S}_P$ (right) ($x$-axis) and generalization gap ($y$-axis). Supplement to Figure 4 in Section 3.1. SGD with a constant learning rate. **No iterate averaging**, and therefore, high randomness in the final state. Same procedures (and models) as in Figure 2. A larger learning rate makes inconsistency larger, but instability is mostly unaffected.

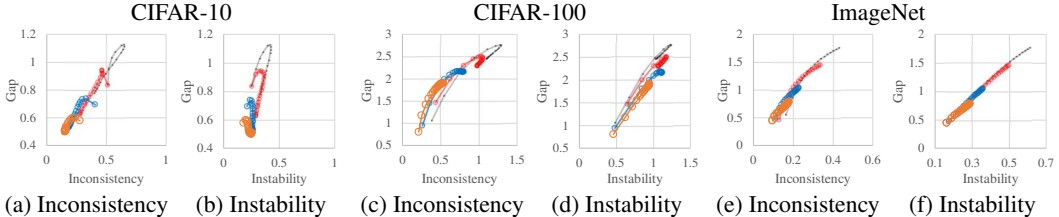

(a) Inconsistency   (b) Instability   (c) Inconsistency   (d) Instability   (e) Inconsistency   (f) Instability

Figure 12: Inconsistency $\mathcal{C}_P$ (left) and instability $\mathcal{S}_P$ (right) ($x$-axis) and generalization gap ($y$-axis). SGD with a constant learning rate **with iterate averaging**, and therefore, low randomness in the final state. Same procedures (and models) as in Figure 1. Both inconsistency and instability are predictive of generalization gap across the learning rates.

1. *Choose a network architecture and the size of training set*. First, for each dataset, we chose a network architecture and the size of the training set. The training set was required to be smaller than the official set so that disjoint training sets can be obtained for estimating instability. For the network architecture, we chose relatively small residual nets (WRN-28-2 for CIFAR-10/100 and ResNet-50 for ImageNet) to reduce the computational burden.

2. *Choose a seed procedure*. Next, for each dataset, we chose a procedure that performs reasonably well with the chosen network architecture and the size of training data, and we call this procedure a *seed procedure*. This was done by referring to the previous studies [32, 10] and performing some tuning on the development data considering that the training data is smaller than in [32, 10]. This step was for making sure to include high-performing (and so practically interesting) procedures in our empirical study.

3. *Make core procedures from the seed procedure*. For each dataset, we made *core procedures* from the seed procedure by varying the learning rate, training length, and the presence/absence of iterate averaging. Table 6 shows the resulting core procedures.

4. *Diversify by changing an attribute*. To make the procedures more diverse, for each dataset, we generated additional procedures by changing *one* attribute of the core procedure. This was done for all the pairs of the core procedures in Table 6 and the attributes in Table 7.

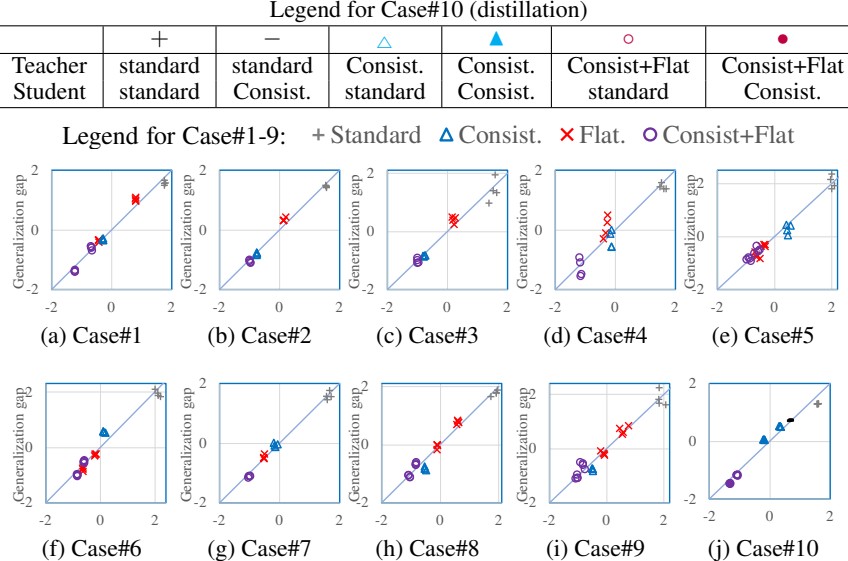

Figure 13: Supplement to Figure 8 in Section 3.2. Inconsistency ($x$-axis) and generalization gap ($y$-axis) for all the 10 cases. All values are standardized so that the average is 0 and the standard deviation is 1.

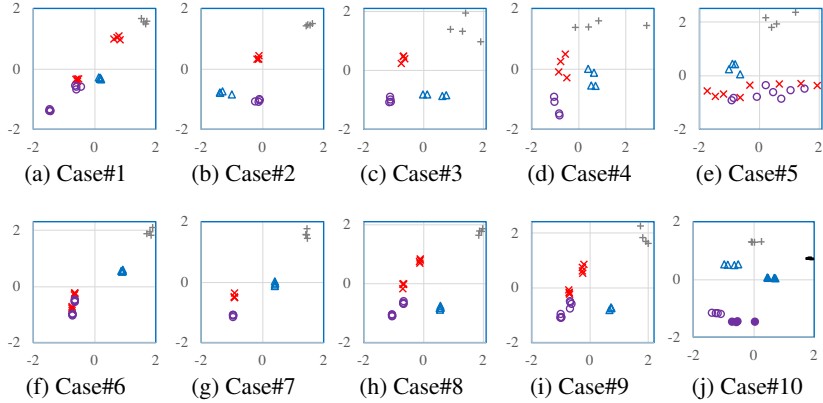

Figure 14: Supplement to Figure 8 in Section 3.2. 1-sharpness ($x$-axis) and generalization gap ($y$-axis) for all the 10 cases. All values are standardized. Same legend as in Figure 13.

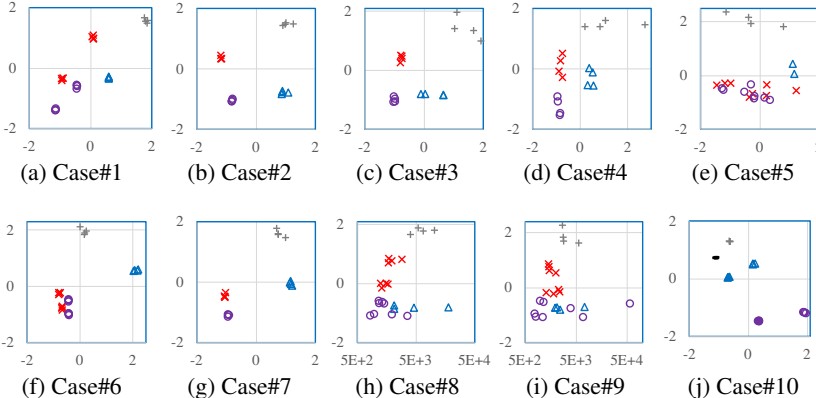

Figure 15: Supplement to Figure 8 in Section 3.2. Hessian ($x$-axis) and generalization gap ($y$-axis) for all the 10 cases. All values are standardized except that for the $x$-axis of Case#8 and 9, non-standardized values are shown in the log-scale for better readability. Same legend as in Figure 13.

Table 6: Core training procedures of the experiments in Section 3.1. The core training procedures consist of the exhaustive combinations of these attributes. The seed procedure attributes are indicated by * when there are multiple values. The optimizer was fixed to SGD with Nesterov momentum 0.9.
† More precisely, { 25, 50, ..., 250, 300, ..., 500, 600, ..., 1000, 1200, ..., 2000 } so that the interval gradually increased from 25 to 200. ‡ After training, a few procedures with very high training loss were excluded from the analysis. The cut-off was 0.3 (CIFAR-10), 2.0 (CIFAR-100), and 3.0 (ImageNet), reflecting the number of classes (10, 100, and 1000).
The choice of the constant scheduling is for starting the empirical study with simpler cases by avoiding the complex effects of decaying learning rates, as mentioned in the main paper; also, we found that in these settings, constant learning rates rival the cosine scheduling as long as iterate averaging is performed.

| | CIFAR-10/100 | ImageNet |
|---|---|---|
| Network | WRN-28-2 | ResNet-50 |
| Training data size | 4K | 120K |
| Learning rate | {0.005, 0.01, 0.025*, 0.05} | {1/64, 1/32, 1/16*, 1/8} |
| Weight decay | 2e-3 | 1e-3 |
| Schedule | Constant | Constant |
| Iterate averaging | {EMA*, None} | {EMA*, None} |
| Epochs | { 25, ..., 1000*, ..., 2000 }†‡ | {10, 20, ..., 200*}‡ |
| Mini-batch size | 64 | 512 |
| Data augmentation | Standard+Cutout | Standard |
| Label smoothing | – | 0.1 |

Table 7: Attributes that were varied for making variations of core procedures. Only one of the attributes was varied at a time.

| | CIFAR-10 | CIFAR-100 | ImageNet |
|---|---|---|---|
| Network | WRN-16-4 | – | – |
| Weight decay | 5e-4 | – | 1e-4 |
| Schedule | Cosine | Cosine | Cosine |
| Mini-batch size | 256 | 256 | – |
| Data augmentation | None | – | – |

Table 8: The values of the fixed attributes of the procedures shown in Figures 1–4 and 6 as well as 11–12. The training length and the learning rate were varied as shown in Table 6. The presence/absence of iterate averaging is indicated in each figure.

| | CIFAR-10 | CIFAR-100 | ImageNet |
|---|---|---|---|
| Network | WRN-28-2 | | ResNet-50 |
| Training data size | 4K | | 120K |
| Weight decay | 2e-3 | | 1e-3 |
| Schedule | Constant | | Constant |
| Mini-batch size | 256 | 64 | 512 |
| Data augmentation | Standard+Cutout | | Standard |
| Label smoothing | – | | 0.1 |

Table 9: The values of the fixed attributes of the procedures shown in Figures 16–19. The training length and the learning rate were varied as shown in Table 6. The presence/absence of iterate averaging is indicated in each figure. The rest of the attributes are the same as in Table 8

| | CIFAR-10 | CIFAR-100 | ImageNet |
|---|---|---|---|
| Weight decay | 5e-4 | 2e-3 | 1e-4 |
| Mini-batch size | 64 | 256 | 512 |

Note that after training, a few procedures with very high training loss were excluded from the analysis (see Table 6 for the cut-off). Right after the model parameter initialization, inconsistency $\mathcal{C}_P$ is obviously not predictive of generalization gap since it is non-zero merely reflecting the initial randomness while generalization gap is zero. Similar effects of initial randomness are expected in the initial phase of training; however, these near random models are not of practical interest. Therefore, we excluded from our analysis.

### C.1.1 SGD with constant learning rates (Figures 1–4 and 6)

In Section 3.1, we first focused on the effects of varying learning rates and training lengths while fixing anything else, with the training procedures that use a constant learning rate with or without iterate averaging. We analyzed all the subsets of the procedures that met this condition, and reported the common trend. That is, when the learning rate is constant, with iterate averaging, $\mathcal{D}_P$ is predictive of generalization gap within and across the learning rates, and without iterate averaging, $\mathcal{D}_P$ is predictive of generalization gap only for the procedures that share the learning rate; moreover, without iterate averaging, larger learning rates cause $\mathcal{D}_P$ to overestimate generalization gap by larger amounts. Figures 1–4 and 6 show one particular subset for each dataset, and Table 8 shows the values of the attributes fixed in these subsets. To demonstrate the generality of the finding, we show the corresponding figures for one more subset for each dataset in Figures 16–19. The values of the fixed attributes in these subsets are shown in Table 9.

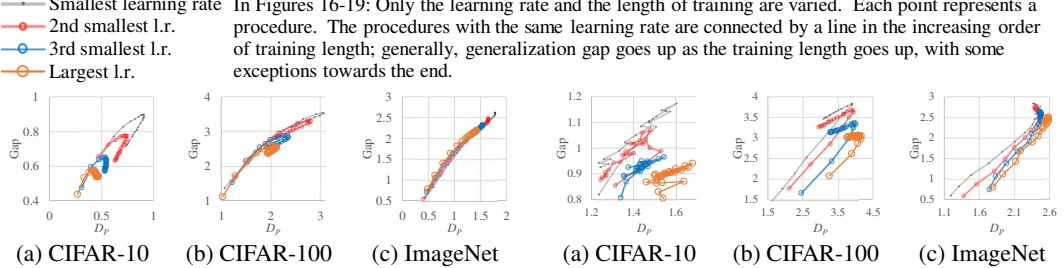

Figure 16: $\mathcal{D}_P$ ($x$-axis) and generalization gap ($y$-axis). SGD with a **constant learning rate** and **iterate averaging**. Only the learning rate and training length were varied as in Figure 1 and the attributes were fixed to the values different from Figure 1; see Table 9 for the fixed values. As in Figure 1, a positive correlation is observed between $\mathcal{D}_P$ and generalization gap.

Figure 17: $\mathcal{D}_P$ ($x$-axis) and generalization gap ($y$-axis). **Constant learning rates. No iterate averaging**. Only the learning rate and training length were varied as in Figure 2 and the attributes were fixed to the values different from Figure 2; see Table 9 for the fixed the values. As in Figure 2, $\mathcal{D}_P$ is predictive of generalization gap for the procedures that share the learning rate, but not clear otherwise.

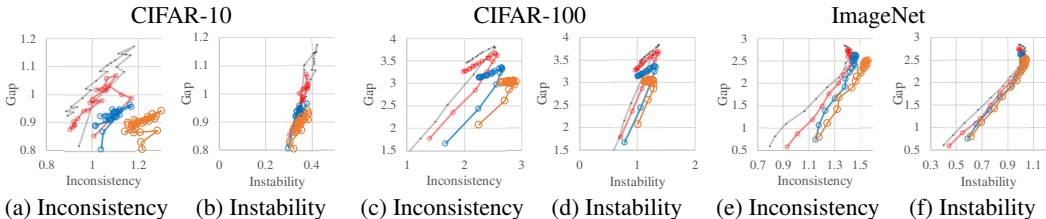

Figure 18: Inconsistency $\mathcal{C}_P$ (left) and instability $\mathcal{S}_P$ (right) ($x$-axis) and generalization gap ($y$-axis). Same procedures (and models) as in Figure 17 (**no iterate averaging**). As in Figures 4 and 11 (also no iterate averaging), a larger learning rate makes inconsistency larger, but instability is mostly unaffected.

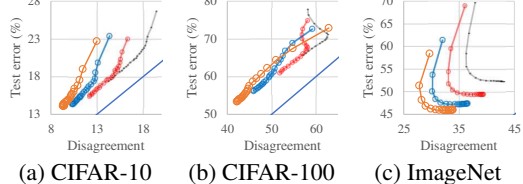

Figure 19: Disagreement ($x$-axis) and test error ($y$-axis). Same models and legend as in Fig 16.

### C.1.2 Procedures with low final randomness (Figures 5 and 7)

The procedures shown in Figures 5 and 7 are subsets (three subsets for three datasets) of all the procedures (the core procedures in Table 6 times the attribute changes in Table 7). The subsets consist of the procedures with either iterate averaging or a vanishing learning rate (i.e., going to zero) so

that they meet the condition of low final randomness. These subsets include those with the cosine learning rate schedule. With the cosine schedule, we were interested in not only letting the learning rate go to zero but also stopping the training before the learning rate reaches zero and setting the final model to be the iterate averaging (EMA) at that point, which is well known to be useful (e.g., [32]). Therefore, we trained the models with the cosine schedule for { 250, 500, 1000, 2000 } epochs (CIFAR-10/100) or 200 epochs (ImageNet), and saved the iterate averaging of the models with the interval of one tenth (CIFAR-10/100) or one twentieth (ImageNet) of the entire epochs.

## C.2 Details of the experiments in Section 3.2

Section 3.2 studied inconsistency in comparison with sharpness in the settings where these two quantities are reduced by algorithms. The training algorithm with consistency encouragement (co-distillation) is summarized in Algorithm 1. These experiments were designed to study

- practical models trained on full-size training data, and
- diverse models resulting from diverse training settings,

in the situation where algorithms are compared after basic tuning is done, rather than the hyperparameter tuning-like situation in Section 3.1.

### C.2.1 Training of the models

Table 10: Basic settings shared by all the models for each case (Case#1–7,10; images)

| Training type | From scratch | | | | Fine-tuning | Distillation |
|---|---|---|---|---|---|---|
| Dataset | ImageNet | | Food101 | CIFAR10 | Cars / Dogs | Food101 |
| Network | ResNet50 | ViT | ViT / Mixer | WRN28-2 | EN-B0 | ResNet-18 |
| Batch size | 512 | 4096 | 512 | 64 | 256 | 512 |
| Epochs | 100 | 300 | 200 / 100 | – | – | 400 |
| Update steps | – | – | – | 500K | 4K / 2K | – |
| Warmup steps | 0 | 10K | 0 | 0 | 0 | 0 |
| Learning rate | 0.125 | 3e-3 | 3e-3 | 0.03 | 0.1 | 0.125 |
| Schedule | Cosine | | Linear/Cosine | Cosine | Constant | Cosine |
| Optimizer | Momentum | AdamW | AdamW | Momentum | Momentum | Momentum |
| Weight decay | 1e-4 | 0.3 | 0.3 | 5e-4 | 1e-5 | 1e-3 |
| Label smooth | 0.1 | 0 | 0.1 | 0 | 0 | 0 |
| Iterate averaging | – | – | – | EMA | EMA | – |
| Gradient clipping | – | 1.0 | 1.0 | – | 20.0 | – |
| Data augment | Standard | | | Cutout | Standard | |
| Reference | [10] | [5] | [5] | [10],[32] | [10],[33] | [10] |
| Case# | 1 | 2 | 3 / 4 | 5 | 6 / 7 | 10 |

'Momentum': SGD with Nesterov momentum 0.9.

This section describes the experiments for producing the models used in Section 3.2.

**Basic settings** Within each of the 10 cases, we used the same basic setting for all, and these shared basic settings were adopted/adapted from the previous studies when possible. Tables 10 and 11

Table 11: Basic settings shared by all the models for each case (Case#8–9; text). Hyperparameters for Case#8–9 (text) basically followed the RoBERTa paper [26]. The learning rate schedule was equivalent to early stopping of 10-epoch linear schedule after 4 epochs. Although it appears that [26] tuned when to stop for each run, we used the same number of epochs for all. Iterate averaging is our addition, which consistently improved performance.

| | |
|---|---|
| Initial learning rate $\eta_0$ | 1e-5 |
| Learning rate schedule | Linear from $\eta_0$ to $0.6\eta_0$ |
| Epochs | 4 |
| Batch size | 32 |
| Optimizer | AdamW ($\beta_1$=0.9, $\beta_2$=0.98, $\epsilon$=1e-6) |
| Weight decay | 0.1 |
| Iterate averaging | EMA with momentum 0.999 |

describe the basic settings and the previous studies that were referred to. Some changes to the previous settings were made for efficiency in our computing environment (no TPU); e.g., for Case#1, we changed the batch size from 4096 to 512 and accordingly the learning rate from 1 to 0.125. When adapting the previous settings to new datasets, minimal tuning was done for obtaining reasonable performance, e.g., for Cases#3 and 4, we changed batch size from 4096 to 512 and kept the learning rate without change as it performed better.

For CIFAR-10, following [32], we let the learning rate decay to $0.2\eta_0$ instead of 0 and set the final model to the EMA of the models with momentum 0.999. For Cases#6–7 (fine-tuning), we used a constant learning rate and used the EMA of the models with momentum 0.999 as the final model, which we found produced reasonable performance with faster training. Cases#6–7 fine-tuned the publicly available EfficientNet-B0[2] pretrained with ImageNet by [33]. The dropout rate was set to 0.1 for Case#3, and the stochastic depth drop rate was set to 0.1 for Case#4. The teacher models for Case#10 (distillation) were ensembles of ResNet-18 trained with label smoothing 0.1 for 200 epochs with the same basic setting as the student models (Table 10) otherwise.

For CIFAR-10, the standard data augmentation (shift and horizontal flip) and Cutout [8] were applied. For the other image datasets, only the standard data augmentation (random crop with distortion and random horizontal flip) was applied; the resolution was $224\times224$.

Table 12: Hyperparameters for SAM.

| Case# | 1 | 2 | 3 | 4 | 5 | 6 | 7 | 8,9 | 10 |
|---|---|---|---|---|---|---|---|---|---|
| $m$-sharpness | 128 | 256 | 32 | 32 | 32 | 16 | 16 | 2 | 128 |
| $\rho$ | 0.05,0.1 | 0.05 | 0.1 | 0.1 | 0.1,0.2 | 0.1,0.2 | 0.1 | 0.005,0.01 | 0.1 |

**Hyperparameters for SAM** There are two values that affect the performance of SAM, $m$ for $m$-sharpness and the diameter of the neighborhood $\rho$. Their values are shown in Table 12. [10] found that smaller $m$ performs better. However, a smaller $m$ can be less efficient as it can reduce the degree of parallelism, depending on the hardware configuration. We made $m$ no greater than the reference study in most cases, but for practical feasibility we made it larger for Case#2. $\rho$ was either set according to the reference when possible or chosen on the development data otherwise, from $\{0.05, 0.1, 0.2\}$ for images and from $\{0.002, 0.005, 0.01, 0.02, 0.05, 0.1\}$ for texts. For some cases (typically those with less computational burden), we trained the models for one additional value of $\rho$ to have more data points.

**Hyperparameters for the inconsistency penalty term** The weight of the inconsistency penalty term for encouraging consistency was fixed to 1.

**Number of the models** For each training procedure (identified by the training objective within each case), we obtained 4 models trained with 4 distinct random sequences. Cases#1–9 consisted of either 4 or 6 procedures depending on the number of the values chosen for $\rho$ for SAM. Case#10 (distillation) consisted of 6 procedures[3], resulting from combining the choice of the training objectives for the teacher and the choice for the student. In total, we had 52 procedures and 208 models.

### C.2.2 Estimation of the model-wise inconsistency and sharpness in Section 3.2

When the expectation over the training set was estimated, for a large training set such as ImageNet, 20K data points were sampled for this purpose. As described above, we had 4 models for each of the training procedures. For the procedure *without* encouragement of low inconsistency, the expectation of the divergence of each model was estimated by taking the average of the divergence from the three other models. As for the procedure *with* encouragement of low inconsistency, the four models were obtained from two runs as each run produced two models, and so when averaging for estimating inconsistency, we excluded the divergence between the models from the same run due to their dependency.

---

[2] https://github.com/google-research/sam

[3] Although the number of all possible combinations is 16, considering the balance with other cases, we chose to experiment with the following: teacher { 'Standard', 'Consist.', 'Consist+Flat' } × student { 'Standard', 'Consist.' }

**Algorithm 1:** Training with consistency encouragement. Co-distillation (named by [1], closely related to deep mutual learning [42]). Without additional unlabeled data.

**Input & Notation**: Labeled set $Z_n$, $\beta$ (default: 1), learning rate $\eta$. Let $\phi$ be loss, and let $f(\theta, x)$ be the model output in the form of probability estimate.

1   Sample $\theta^{\mathrm{a}}$ and $\theta^{\mathrm{b}}$ from the initial distribution.
2   **for** $t = 1, \ldots, T$ **do**
3      Sample two labeled mini-batches $B^{\mathrm{a}}$ and $B^{\mathrm{b}}$ from $Z_n$.
4      $\theta^{\mathrm{a}} \leftarrow \theta^{\mathrm{a}} - \eta_t \nabla_{\theta^{\mathrm{a}}} \left[ \frac{1}{|B^{\mathrm{a}}|} \sum_{(x,y) \in B^{\mathrm{a}}} \phi(f(\theta^{\mathrm{a}}, x), y) + \beta \frac{1}{|B^{\mathrm{a}}|} \sum_{(x,\cdot) \in B^{\mathrm{a}}} \mathrm{KL}(f(\theta^{\mathrm{b}}, x) || f(\theta^{\mathrm{a}}, x)) \right]$
5      $\theta^{\mathrm{b}} \leftarrow \theta^{\mathrm{b}} - \eta_t \nabla_{\theta^{\mathrm{b}}} \left[ \frac{1}{|B^{\mathrm{b}}|} \sum_{(x,y) \in B^{\mathrm{b}}} \phi(f(\theta^{\mathrm{b}}, x), y) + \beta \frac{1}{|B^{\mathrm{b}}|} \sum_{(x,\cdot) \in B^{\mathrm{b}}} \mathrm{KL}(f(\theta^{\mathrm{a}}, x) || f(\theta^{\mathrm{b}}, x)) \right]$

1-sharpness requires $\rho$ (the diameter of the local region) as input. We set it to the best value for SAM.

### C.3   Details of the experiments in Sections 3.3

The optimizer was SGD with Nesterov momentum 0.9.

Table 13: Supplement to Figure 9 (a). Error rate (%) and inconsistency of ensembles in comparison with non-ensemble models (+). Food101, ResNet-18. The average and standard deviation of 4 are shown. Ensemble reduces test error and inconsistency.

| | Training method | Test error(%) | inconsistency |
|---|---|---|---|
| + | Non-ensemble | $17.09_{\pm 0.20}$ | $0.30_{\pm 0.001}$ |
| * | Ensemble of standard models | $14.99_{\pm 0.05}$ | $0.14_{\pm 0.001}$ |
| • | Ensembles of Consist. models | $14.07_{\pm 0.08}$ | $0.10_{\pm 0.001}$ |

#### C.3.1   Ensemble experiments (Figure 9 (a))

The models used in the ensemble experiments were ResNet-18 trained for 200 epochs with label smoothing 0.1 with the basic setting of Case#10 in Table 10 otherwise. Table 13 shows the standard deviation of the values presented in Figure 9 (a). The ensembles also served as the teachers in the distillation experiments.

#### C.3.2   Distillation experiments (Figure 9 (b))

The student models were ResNet-18 trained for 200 epochs with the basic setting of Case#10 of Table 10 otherwise. The teachers for Figure 9(b) (left) were ResNet-18 ensemble models trained as described in C.3.1. The teachers for Figure 9(b) (middle) were ResNet-50 ensemble models trained similarly. The teachers for Figure 9(b) (right) were EfficientNet-B0 models obtained by fine-tuning the public ImageNet-trained model (footnote 2); fine-tuning was done with encouragement of consistency and flatness (with $\rho$=0.1) with batch size 512, weight decay 1e-5, the initial learning rate 0.1 with cosign scheduling, gradient clipping 20, and 20K updates. Table 14 shows the standard deviations of the values plotted in Figure 9 (b).

Table 14: Supplement to Figure 9 (b). Test error (%) and inconsistency of distilled-models in comparison with standard models (+). The average and standard deviation of 4 are shown.

| | Test error (%) | | | inconsistency | | |
|---|---|---|---|---|---|---|
| | left | middle | right | left | middle | right |
| + | | $17.09_{\pm 0.20}$ | | | $0.30_{\pm 0.001}$ | |
| △ | $15.74_{\pm 0.09}$ | $14.95_{\pm 0.17}$ | $14.61_{\pm 0.14}$ | $0.19_{\pm 0.001}$ | $0.21_{\pm 0.001}$ | $0.22_{\pm 0.002}$ |
| ▲ | $14.99_{\pm 0.07}$ | $14.28_{\pm 0.11}$ | $13.89_{\pm 0.08}$ | $0.14_{\pm 0.001}$ | $0.15_{\pm 0.001}$ | $0.14_{\pm 0.002}$ |
| • | $14.41_{\pm 0.05}$ | $13.63_{\pm 0.09}$ | $13.06_{\pm 0.06}$ | $0.10_{\pm 0.001}$ | $0.12_{\pm 0.001}$ | $0.08_{\pm 0.001}$ |

**Algorithm 2:** Our semi-supervised variant of co-distillation.

**Input & Notation**: Labeled set $Z_n$, unlabeled set $U$, $\beta$ (default: 1), $\tau$ (default: 0.5), learning rate $\eta$, momentum of EMA (default: 0.999). Let $\phi$ be loss, and let $f(\theta, x)$ be the model output in the form of probability estimate.

1   Initialize $\theta^{\mathrm{a}}$, $\theta^{\mathrm{b}}$, $\bar{\theta}^{\mathrm{a}}$, and $\bar{\theta}^{\mathrm{b}}$.

2   **for** $t = 1, \ldots, T$ **do**

3     Sample labeled mini-batches $B^{\mathrm{a}}$ and $B^{\mathrm{b}}$ from $Z_n$, and sample an unlabeled mini-batch $B_U$ from $U$.

4     Let $\psi(\theta, \bar{\theta}, x) = \beta \mathbb{I}(\max_i f(\bar{\theta}; x)[i] > \tau)\mathrm{KL}(f(\bar{\theta}, x) || f(\theta, x))$ where $\mathbb{I}$ is the indicator function.

5     $\theta^{\mathrm{a}} \leftarrow \theta^{\mathrm{a}} - \eta_t \nabla_{\theta^{\mathrm{a}}} \left[ \frac{1}{|B^{\mathrm{a}}|} \sum_{(x,y) \in B^{\mathrm{a}}} \phi(f(\theta^{\mathrm{a}}, x), y) + \frac{1}{|B_U|} \sum_{x \in B_U} \psi(\theta^{\mathrm{a}}, \bar{\theta}^{\mathrm{b}}, x) \right]$

6     $\theta^{\mathrm{b}} \leftarrow \theta^{\mathrm{b}} - \eta_t \nabla_{\theta^{\mathrm{b}}} \left[ \frac{1}{|B^{\mathrm{b}}|} \sum_{(x,y) \in B^{\mathrm{b}}} \phi(f(\theta^{\mathrm{b}}, x), y) + \frac{1}{|B_U|} \sum_{x \in B_U} \psi(\theta^{\mathrm{b}}, \bar{\theta}^{\mathrm{a}}, x) \right]$

7     $\bar{\theta}^{\mathrm{a}}$ and $\bar{\theta}^{\mathrm{b}}$ keep the EMA of $\theta^{\mathrm{a}}$ and $\theta^{\mathrm{b}}$, with momentum $\mu$, respectively.

### C.3.3   Semi-supervised experiments reported in Table 4

The unlabeled data experiments reported in Table 4 used our modification of Algorithm 1, taylored for use of unlabeled data, and it is summarized in Algorithm 2. It differs from Algorithm 1 in two ways. First, to compute the inconsistency penalty term, the model output is compared with that of the exponential moving average (EMA) of the other model, reminiscent of Mean Teacher [34]. The output of the EMA model during the training typically has higher confidence (or lower entropy) than the model itself, and so taking the KL divergence against EMA of the other model serves the purpose of sharpening the pseudo labels and reducing the entropy on unlabeled data. Second, we adopted the masked divergence from *Unsupervised Data Augmentation (UDA)* [37], which masks (i.e., ignores) the unlabeled instances for which the confidence level of the other model is less than threshold. These changes are effective in the semi-supervised setting (but not in the supervised setting) for preventing the models from getting stuck in a high-entropy region.

In the experiments reported in Table 4, we penalized inconsistency between the model outputs of two training instances of *Unsupervised Data Augmentation (UDA)* [37], using Algorithm 2, by replacing loss $\phi(\theta, x)$ with the UDA objective. The UDA objective penalizes discrepancies between the model outputs for two different data representations (a strongly augmented one and a weakly augmented one) on the unlabeled data (the *UDA penalty*). The inconsistency penalty term of Algorithm 2 also uses unlabeled data, and for this purpose, we used a strongly augmented unlabeled batch sampled independently of those for the UDA penalty. UDA 'sharpens' the model output on the weakly augmented data by scaling the logits, which serves as pseudo labels for unlabeled data, and the degree of sharpening is a tuning parameter. However, we tested UDA without sharpening and obtained better performance on the development data (held-out 5K data points) than reported in [32], and so we decided to use UDA without sharpening for our UDA+'Consist.' experiments. We obtained test error 3.95% on the average of 5 independent runs, which is better than 4.33% of UDA alone and 4.25% of FixMatch. Note that each of the 5 runs used a different fold of 4K examples as was done in the previous studies.

Following [32], we used labeled batch size 64, weight decay 5e-4, and updated the weights 500K times with the cosine learning rate schedule decaying from 0.03 to 0.2×0.03. We set the final model to the average of the last 5% iterates (i.e., the last 25K snapshots of model parameters). We used these same basic settings for all (UDA, FixMatch, and UDA+'Consist.'). The unlabeled batch size for testing UDA and FixMatch was 64×7, as in [32]. For computing each of the two penalty terms for UDA+'Consist.', we set the unlabeled batch size to 64×4, which is approximately one half of that for UDA and FixMatch. We made it one half so that the total number of unlabeled data points used by each model (64×4 for the UDA penalty plus 64×4 for the inconsistency penalty) becomes similar to that of UDA or FixMatch. RandAugment with the same modification as described in the FixMatch study was used as strong augmentation, and the standard data augmentation (shift and flip) was used as weak augmentation. The threshold for masking was set to 0.5 for both the UDA penalty and the inconsistency penalty and the weights of the both penalties were set to 1. Note that we fixed the

weights of penalties to 1, and only tuned the threshold for masking by selecting from $\{0, 0.5, 0.7\}$ on the development data (5K examples).

### C.3.4 Fine-tuning experiments in Table 5

The EfficientNet-B4 (EN-B4) fine-tuning experiments reported in Table 5 were done with weight decay 1e-5 (following [10]), batch size 256 and number of updates 20K following the previous studies. We set the learning rate to 0.1 and performed gradient clipping with size 20 to deal with a sudden surge of the gradient size. The diameter of the local region $\rho$ for SAM was chosen from $\{0.05, 0.1, 0.2\}$ based on the performance of SAM on the development data, and the same chosen value 0.2 was used for both SAM and SAM+'Consist.' Following the EN-B7 experiments of [10], the value $m$ for $m$-sharpness was set to 16. Since SAM+'Consist.' is approximately twice as expensive as SAM as a result of training two models, we also tested SAM with 40K updates (20K×2) and found that it did not improve performance. We note that our baseline EN-B4 SAM performance is better than the EN-B7 SAM performance of [10]. This is due to the difference in the basic setting. In [10], EN-B7 was fine-tuned with a larger batch size 1024 with a smaller learning rate 0.016 while the batch normalization statistics was fixed to the pre-trained statistics. Our basic setting allowed a model to go farther away from the initial pre-trained model. Also note that we experimented with smaller EN-B4 instead of EN-B7 due to resource constraints.

## D   Additional information

### D.1   Additional correlation analyses using the framework of Jiang et al. (2020)

This section reports on the additional correlation analysis using the rigorous framework of Jiang et al. (2020) [18] and shows that the results are consistent with the results in the main paper and the previous work. The analysis uses correlation metrics proposed by [18], which seek to mitigate the effect of what [18] calls *spurious correlations* that do not reflect causal relationships with generalization. For completeness, we briefly describe below these metrics, and [18] should be referred to for more details and justification.

**Notation**   In this section, we write $\pi$ for a training procedure (or equivalently, a combination of hyperparameters including the network architecture, the data augmentation method, and so forth). Let $g(\pi)$ be the generalization gap of $\pi$, and let $\mu(\pi)$ be the quantity of interest such as inconsistency or disagreement.

**Ranking-based**   Let $\mathcal{T}$ be the set of the corresponding pairs of generalization gap and the quantity of interest to be considered: $\mathcal{T} := \cup_\pi \{(\mu(\pi), g(\pi))\}$. Then the standard Kendall's ranking coefficient $\tau$ of $\mathcal{T}$ can be expressed as:

$$\tau(\mathcal{T}) := \frac{1}{|\mathcal{T}|(|\mathcal{T}|-1)} \sum_{(\mu_1, g_1) \in \mathcal{T}} \sum_{(\mu_2, g_2) \in \mathcal{T} \setminus (\mu_1, g_1)} \text{sign}(\mu_1 - \mu_2)\text{sign}(g_1 - g_2)$$

[18] defines *granulated Kendall's coefficient* $\Psi$ as:

$$\Psi := \frac{1}{n} \sum_{i=1}^n \psi_i, \ \psi_i := \frac{1}{m_i} \sum_{\pi_1 \in \Pi_1} \cdots \sum_{\pi_{i-1} \in \Pi_{i-1}} \sum_{\pi_{i+1} \in \Pi_{i+1}} \cdots \sum_{\pi_n \in \Pi_n} \tau\left(\cup_{\pi_i \in \Pi_i}(\mu(\pi), g(\pi))\right) \quad (6)$$

where $\pi_i$ is the $i$-th hyperparameter so that $\pi = (\pi_1, \pi_2, \cdots, \pi_n)$, $\Pi_i$ is the set of all possible values for the $i$-th hyperparameter, and $m_i := |\Pi_1 \times \cdots \times \Pi_{i-1} \times \Pi_{i+1} \times \cdots \times \Pi_n|$.

**Mutual information-based**   Define $V_g(\pi, \pi') := \text{sign}(g(\pi) - g(\pi'))$, and similarly define $V_\mu(\pi, \pi') := \text{sign}(\mu(\pi) - \mu(\pi'))$. Let $U_S$ be a random variable representing the values of the hyperparameter types in $S$ (e.g., $S = \{$ learning rate, batch size $\}$). Then $I(V_\mu, V_g|U_S)$, the conditional mutual information between $\mu$ and $g$ given the set $S$ of hyperparameter types, and the

Table 15: Correlation scores of inconsistency and disagreement. For the training procedures with low final randomness (as in Figure 5), model-wise quantities (one model per procedure) were analyzed. (a)–(c) differ in the restriction on training loss; (b) and (c) exclude the models with high training loss while (a) does not. The average and standard deviation of 4 independent runs (that use 4 distinct subsamples of training sets as training data and distinct random seeds) are shown. **Correlation scores**: Two types of mutual information-based scores ('$\mathcal{K}$' as in (7) and '$|S|{=}0$': $\frac{I(V_\mu, V_g | U_S)}{H(V_g | U_S)}$ with $|S|{=}0$) and two types of Kendall's rank-correlation coefficient-based scores ($\Psi$ as in (6) and overall $\tau$). A larger number indicates a higher correlation. The highest numbers are highlighted. **Tested quantities**: 'Inconsist.': Inconsistency, 'Disagree.': Disagreement, 'Random' (baseline): random numbers drawn from the normal distribution, 'Canonical' (baseline): a slight extension of the canonical ordering in [18]; it heuristically determines the order of two procedures by preferring smaller batch size, larger weight decay, larger learning rate, and presence of data augmentation (which are considered to be associated with better generalization) by adding one point for each and breaking ties randomly. **Target quantities**: Generalization gap (test loss minus training loss) as defined in the main paper, test error, and test error minus training error. **Observation**: Inconsistency correlates well to generalization gap, and disagreement correlates well to test error. With more aggressive exclusion of the models with high training loss (going from (a) to (c)), the correlation of disagreement to generalization gap improves and approaches that of inconsistency.

(a) **No restriction** on training loss

|  | CIFAR-10 | | | | CIFAR-100 | | | | ImageNet | | | |
|---|---|---|---|---|---|---|---|---|---|---|---|---|
|  | $\mathcal{K}$ | $|S|{=}0$ | $\Psi$ | $\tau$ | $\mathcal{K}$ | $|S|{=}0$ | $\Psi$ | $\tau$ | $\mathcal{K}$ | $|S|{=}0$ | $\Psi$ | $\tau$ |
| Correlation to generalization gap ('test loss - training loss') | | | | | | | | | | | | |
| Inconsist. | $\mathbf{0.23}_{\pm.01}$ | $\mathbf{0.38}_{\pm.01}$ | $\mathbf{0.60}_{\pm.05}$ | $\mathbf{0.69}_{\pm.01}$ | $\mathbf{0.51}_{\pm.01}$ | $\mathbf{0.55}_{\pm.01}$ | $\mathbf{0.68}_{\pm.06}$ | $\mathbf{0.81}_{\pm.01}$ | $\mathbf{0.75}_{\pm.01}$ | $\mathbf{0.75}_{\pm.01}$ | $\mathbf{0.73}_{\pm.17}$ | $\mathbf{0.92}_{\pm.00}$ |
| Disagree. | $0.14_{\pm.01}$ | $0.26_{\pm.01}$ | $0.54_{\pm.05}$ | $0.58_{\pm.01}$ | $0.11_{\pm.01}$ | $0.11_{\pm.01}$ | $0.48_{\pm.06}$ | $0.39_{\pm.01}$ | $0.11_{\pm.01}$ | $0.16_{\pm.01}$ | $0.53_{\pm.07}$ | $0.47_{\pm.01}$ |
| Random | $0.00_{\pm.00}$ | $0.00_{\pm.00}$ | $-0.02_{\pm.04}$ | $0.01_{\pm.03}$ | $0.00_{\pm.00}$ | $0.00_{\pm.00}$ | $-0.01_{\pm.07}$ | $0.02_{\pm.02}$ | $0.00_{\pm.00}$ | $0.01_{\pm.00}$ | $-0.00_{\pm.11}$ | $-0.04_{\pm.08}$ |
| Canonical | $0.01_{\pm.00}$ | $0.10_{\pm.00}$ | $0.32_{\pm.03}$ | $0.37_{\pm.01}$ | $0.00_{\pm.00}$ | $0.09_{\pm.00}$ | $0.14_{\pm.05}$ | $0.36_{\pm.01}$ | $0.00_{\pm.00}$ | $0.07_{\pm.00}$ | $0.36_{\pm.05}$ | $0.32_{\pm.00}$ |
| Correlation to test error | | | | | | | | | | | | |
| Inconsist. | $0.08_{\pm.01}$ | $0.16_{\pm.01}$ | $0.51_{\pm.04}$ | $0.46_{\pm.01}$ | $0.00_{\pm.00}$ | $0.01_{\pm.00}$ | $0.33_{\pm.01}$ | $0.14_{\pm.01}$ | $0.03_{\pm.00}$ | $0.03_{\pm.00}$ | $0.20_{\pm.03}$ | $-0.20_{\pm.00}$ |
| Disagree. | $\mathbf{0.31}_{\pm.01}$ | $\mathbf{0.37}_{\pm.00}$ | $\mathbf{0.58}_{\pm.05}$ | $\mathbf{0.68}_{\pm.00}$ | $\mathbf{0.19}_{\pm.01}$ | $\mathbf{0.31}_{\pm.02}$ | $\mathbf{0.57}_{\pm.05}$ | $\mathbf{0.63}_{\pm.01}$ | $\mathbf{0.04}_{\pm.00}$ | $\mathbf{0.05}_{\pm.00}$ | $\mathbf{0.30}_{\pm.07}$ | $\mathbf{0.25}_{\pm.01}$ |
| Random | $0.00_{\pm.00}$ | $0.00_{\pm.00}$ | $0.02_{\pm.03}$ | $0.00_{\pm.01}$ | $0.00_{\pm.00}$ | $0.00_{\pm.00}$ | $-0.05_{\pm.07}$ | $-0.00_{\pm.04}$ | $0.00_{\pm.00}$ | $0.00_{\pm.00}$ | $-0.07_{\pm.08}$ | $0.01_{\pm.04}$ |
| Canonical | $0.01_{\pm.00}$ | $0.03_{\pm.00}$ | $0.15_{\pm.02}$ | $0.20_{\pm.01}$ | $0.00_{\pm.00}$ | $0.19_{\pm.01}$ | $0.31_{\pm.15}$ | $0.49_{\pm.01}$ | $0.00_{\pm.00}$ | $0.03_{\pm.00}$ | $0.08_{\pm.03}$ | $0.18_{\pm.00}$ |
| Correlation to 'test error - training error' | | | | | | | | | | | | |
| Inconsist. | $0.11_{\pm.00}$ | $0.25_{\pm.01}$ | $0.53_{\pm.04}$ | $0.57_{\pm.01}$ | $\mathbf{0.41}_{\pm.01}$ | $\mathbf{0.48}_{\pm.01}$ | $\mathbf{0.60}_{\pm.02}$ | $\mathbf{0.77}_{\pm.01}$ | $\mathbf{0.73}_{\pm.01}$ | $\mathbf{0.73}_{\pm.00}$ | $\mathbf{0.91}_{\pm.03}$ | $\mathbf{0.91}_{\pm.00}$ |
| Disagree. | $\mathbf{0.16}_{\pm.01}$ | $\mathbf{0.28}_{\pm.01}$ | $\mathbf{0.54}_{\pm.07}$ | $\mathbf{0.60}_{\pm.01}$ | $0.12_{\pm.01}$ | $0.12_{\pm.01}$ | $0.45_{\pm.05}$ | $0.40_{\pm.01}$ | $0.12_{\pm.01}$ | $0.16_{\pm.01}$ | $0.52_{\pm.06}$ | $0.46_{\pm.01}$ |
| Random | $0.00_{\pm.00}$ | $0.00_{\pm.00}$ | $0.03_{\pm.04}$ | $0.00_{\pm.02}$ | $0.00_{\pm.00}$ | $0.00_{\pm.00}$ | $0.01_{\pm.07}$ | $-0.01_{\pm.03}$ | $0.00_{\pm.00}$ | $0.00_{\pm.00}$ | $0.06_{\pm.17}$ | $-0.03_{\pm.07}$ |
| Canonical | $0.01_{\pm.00}$ | $0.07_{\pm.00}$ | $0.31_{\pm.06}$ | $0.31_{\pm.01}$ | $0.00_{\pm.00}$ | $0.09_{\pm.00}$ | $0.21_{\pm.09}$ | $0.35_{\pm.01}$ | $0.00_{\pm.00}$ | $0.07_{\pm.00}$ | $0.24_{\pm.02}$ | $0.32_{\pm.00}$ |

(b) Excluding the models with very high training loss, as described in Appendix C.1 and Table 6.

|  | CIFAR-10 | | | | CIFAR-100 | | | | ImageNet | | | |
|---|---|---|---|---|---|---|---|---|---|---|---|---|
|  | $\mathcal{K}$ | $|S|{=}0$ | $\Psi$ | $\tau$ | $\mathcal{K}$ | $|S|{=}0$ | $\Psi$ | $\tau$ | $\mathcal{K}$ | $|S|{=}0$ | $\Psi$ | $\tau$ |
| Correlation to generalization gap ('test loss - training loss') | | | | | | | | | | | | |
| Inconsist. | $\mathbf{0.37}_{\pm.01}$ | $\mathbf{0.57}_{\pm.01}$ | $\mathbf{0.64}_{\pm.05}$ | $\mathbf{0.83}_{\pm.01}$ | $\mathbf{0.47}_{\pm.01}$ | $\mathbf{0.52}_{\pm.01}$ | $\mathbf{0.68}_{\pm.07}$ | $\mathbf{0.79}_{\pm.01}$ | $\mathbf{0.74}_{\pm.01}$ | $\mathbf{0.75}_{\pm.01}$ | $\mathbf{0.74}_{\pm.16}$ | $\mathbf{0.92}_{\pm.00}$ |
| Disagree. | $0.34_{\pm.02}$ | $0.53_{\pm.02}$ | $0.61_{\pm.05}$ | $0.80_{\pm.02}$ | $0.26_{\pm.03}$ | $0.33_{\pm.01}$ | $0.60_{\pm.05}$ | $0.65_{\pm.01}$ | $0.24_{\pm.02}$ | $0.33_{\pm.02}$ | $0.61_{\pm.07}$ | $0.65_{\pm.01}$ |
| Random | $0.00_{\pm.00}$ | $0.00_{\pm.00}$ | $-0.01_{\pm.05}$ | $-0.00_{\pm.03}$ | $0.00_{\pm.00}$ | $0.00_{\pm.00}$ | $0.06_{\pm.06}$ | $-0.01_{\pm.03}$ | $0.00_{\pm.00}$ | $0.00_{\pm.00}$ | $-0.10_{\pm.07}$ | $-0.05_{\pm.07}$ |
| Canonical | $0.02_{\pm.00}$ | $0.15_{\pm.00}$ | $0.28_{\pm.04}$ | $0.44_{\pm.01}$ | $0.00_{\pm.00}$ | $0.21_{\pm.01}$ | $0.23_{\pm.06}$ | $0.52_{\pm.01}$ | $0.00_{\pm.00}$ | $0.10_{\pm.00}$ | $0.46_{\pm.16}$ | $0.38_{\pm.00}$ |
| Correlation to test error | | | | | | | | | | | | |
| Inconsist. | $0.10_{\pm.00}$ | $0.18_{\pm.01}$ | $0.51_{\pm.04}$ | $0.49_{\pm.01}$ | $0.01_{\pm.00}$ | $0.10_{\pm.01}$ | $0.42_{\pm.00}$ | $0.36_{\pm.02}$ | $0.01_{\pm.00}$ | $0.01_{\pm.00}$ | $0.21_{\pm.02}$ | $-0.10_{\pm.00}$ |
| Disagree. | $\mathbf{0.27}_{\pm.00}$ | $\mathbf{0.33}_{\pm.01}$ | $\mathbf{0.57}_{\pm.06}$ | $\mathbf{0.65}_{\pm.00}$ | $\mathbf{0.15}_{\pm.01}$ | $\mathbf{0.27}_{\pm.01}$ | $\mathbf{0.56}_{\pm.05}$ | $\mathbf{0.59}_{\pm.01}$ | $\mathbf{0.02}_{\pm.00}$ | $\mathbf{0.02}_{\pm.00}$ | $\mathbf{0.25}_{\pm.07}$ | $\mathbf{0.16}_{\pm.01}$ |
| Random | $0.00_{\pm.00}$ | $0.00_{\pm.00}$ | $0.03_{\pm.03}$ | $0.01_{\pm.02}$ | $0.00_{\pm.00}$ | $0.00_{\pm.00}$ | $0.04_{\pm.07}$ | $0.02_{\pm.02}$ | $0.00_{\pm.00}$ | $0.00_{\pm.00}$ | $0.04_{\pm.10}$ | $0.01_{\pm.06}$ |
| Canonical | $0.01_{\pm.00}$ | $0.03_{\pm.00}$ | $0.14_{\pm.02}$ | $0.21_{\pm.01}$ | $0.00_{\pm.00}$ | $0.20_{\pm.01}$ | $0.25_{\pm.14}$ | $0.51_{\pm.01}$ | $0.00_{\pm.00}$ | $0.02_{\pm.00}$ | $0.41_{\pm.03}$ | $0.19_{\pm.00}$ |
| Correlation to 'test error - training error' | | | | | | | | | | | | |
| Inconsist. | $0.19_{\pm.01}$ | $0.37_{\pm.01}$ | $0.57_{\pm.04}$ | $0.69_{\pm.01}$ | $\mathbf{0.35}_{\pm.01}$ | $\mathbf{0.44}_{\pm.01}$ | $\mathbf{0.60}_{\pm.02}$ | $\mathbf{0.74}_{\pm.01}$ | $\mathbf{0.72}_{\pm.01}$ | $\mathbf{0.73}_{\pm.00}$ | $\mathbf{0.93}_{\pm.03}$ | $\mathbf{0.91}_{\pm.00}$ |
| Disagree. | $\mathbf{0.35}_{\pm.02}$ | $\mathbf{0.52}_{\pm.02}$ | $\mathbf{0.60}_{\pm.06}$ | $\mathbf{0.80}_{\pm.01}$ | $0.30_{\pm.03}$ | $0.35_{\pm.01}$ | $0.57_{\pm.05}$ | $0.67_{\pm.01}$ | $0.25_{\pm.02}$ | $0.32_{\pm.01}$ | $0.61_{\pm.06}$ | $0.64_{\pm.01}$ |
| Random | $0.00_{\pm.00}$ | $0.00_{\pm.00}$ | $0.01_{\pm.06}$ | $-0.02_{\pm.01}$ | $0.00_{\pm.00}$ | $0.00_{\pm.00}$ | $-0.01_{\pm.08}$ | $0.01_{\pm.03}$ | $0.00_{\pm.00}$ | $0.00_{\pm.00}$ | $-0.02_{\pm.09}$ | $-0.06_{\pm.06}$ |
| Canonical | $0.01_{\pm.00}$ | $0.10_{\pm.00}$ | $0.29_{\pm.03}$ | $0.36_{\pm.01}$ | $0.00_{\pm.00}$ | $0.20_{\pm.01}$ | $0.23_{\pm.06}$ | $0.50_{\pm.01}$ | $0.00_{\pm.00}$ | $0.11_{\pm.00}$ | $0.65_{\pm.03}$ | $0.38_{\pm.00}$ |

(c) Excluding the models with high training loss **more aggressively** with smaller cut-off values (one half of (b))

|  | CIFAR-10 | | | | CIFAR-100 | | | | ImageNet | | | |
|---|---|---|---|---|---|---|---|---|---|---|---|---|
|  | $\mathcal{K}$ | $|S|{=}0$ | $\Psi$ | $\tau$ | $\mathcal{K}$ | $|S|{=}0$ | $\Psi$ | $\tau$ | $\mathcal{K}$ | $|S|{=}0$ | $\Psi$ | $\tau$ |
| Correlation to generalization gap ('test loss - training loss') | | | | | | | | | | | | |
| Inconsist. | $0.36_{\pm.01}$ | $\mathbf{0.57}_{\pm.01}$ | $\mathbf{0.65}_{\pm.05}$ | $\mathbf{0.82}_{\pm.01}$ | $\mathbf{0.44}_{\pm.01}$ | $\mathbf{0.50}_{\pm.01}$ | $\mathbf{0.69}_{\pm.06}$ | $\mathbf{0.78}_{\pm.01}$ | $\mathbf{0.77}_{\pm.01}$ | $\mathbf{0.79}_{\pm.01}$ | $\mathbf{0.74}_{\pm.16}$ | $\mathbf{0.93}_{\pm.00}$ |
| Disagree. | $\mathbf{0.39}_{\pm.01}$ | $\mathbf{0.57}_{\pm.01}$ | $0.62_{\pm.05}$ | $\mathbf{0.82}_{\pm.01}$ | $0.31_{\pm.02}$ | $0.43_{\pm.01}$ | $0.65_{\pm.05}$ | $0.73_{\pm.01}$ | $0.43_{\pm.01}$ | $0.53_{\pm.01}$ | $0.69_{\pm.06}$ | $0.80_{\pm.01}$ |
| Random | $0.00_{\pm.00}$ | $0.00_{\pm.00}$ | $0.02_{\pm.03}$ | $0.00_{\pm.02}$ | $0.00_{\pm.00}$ | $0.00_{\pm.00}$ | $0.03_{\pm.11}$ | $0.04_{\pm.02}$ | $0.00_{\pm.00}$ | $0.00_{\pm.00}$ | $-0.05_{\pm.12}$ | $-0.02_{\pm.03}$ |
| Canonical | $0.02_{\pm.00}$ | $0.14_{\pm.00}$ | $0.36_{\pm.04}$ | $0.44_{\pm.01}$ | $0.00_{\pm.00}$ | $0.27_{\pm.01}$ | $0.23_{\pm.06}$ | $0.60_{\pm.01}$ | $0.00_{\pm.00}$ | $0.13_{\pm.00}$ | $0.44_{\pm.09}$ | $0.42_{\pm.00}$ |
| Correlation to test error | | | | | | | | | | | | |
| Inconsist. | $0.12_{\pm.01}$ | $0.21_{\pm.01}$ | $0.53_{\pm.04}$ | $0.53_{\pm.01}$ | $0.04_{\pm.00}$ | $0.19_{\pm.01}$ | $0.48_{\pm.00}$ | $0.50_{\pm.02}$ | $0.01_{\pm.00}$ | $0.01_{\pm.00}$ | $\mathbf{0.31}_{\pm.02}$ | $0.13_{\pm.00}$ |
| Disagree. | $\mathbf{0.27}_{\pm.00}$ | $\mathbf{0.35}_{\pm.01}$ | $\mathbf{0.58}_{\pm.06}$ | $\mathbf{0.66}_{\pm.01}$ | $\mathbf{0.19}_{\pm.02}$ | $\mathbf{0.34}_{\pm.01}$ | $\mathbf{0.58}_{\pm.06}$ | $\mathbf{0.65}_{\pm.01}$ | $\mathbf{0.05}_{\pm.00}$ | $\mathbf{0.05}_{\pm.00}$ | $0.27_{\pm.08}$ | $\mathbf{0.25}_{\pm.00}$ |
| Random | $0.00_{\pm.00}$ | $0.00_{\pm.00}$ | $-0.01_{\pm.03}$ | $0.01_{\pm.03}$ | $0.00_{\pm.00}$ | $0.00_{\pm.00}$ | $0.01_{\pm.11}$ | $0.01_{\pm.04}$ | $0.00_{\pm.00}$ | $0.00_{\pm.00}$ | $-0.01_{\pm.15}$ | $-0.01_{\pm.01}$ |
| Canonical | $0.01_{\pm.00}$ | $0.03_{\pm.00}$ | $0.25_{\pm.03}$ | $0.22_{\pm.01}$ | $0.00_{\pm.00}$ | $0.24_{\pm.01}$ | $0.25_{\pm.03}$ | $0.56_{\pm.01}$ | $0.00_{\pm.00}$ | $0.06_{\pm.00}$ | $0.20_{\pm.10}$ | $0.29_{\pm.01}$ |
| Correlation to 'test error - training error' | | | | | | | | | | | | |
| Inconsist. | $0.18_{\pm.01}$ | $0.37_{\pm.01}$ | $0.58_{\pm.04}$ | $0.68_{\pm.01}$ | $0.30_{\pm.01}$ | $0.42_{\pm.01}$ | $0.61_{\pm.02}$ | $0.72_{\pm.01}$ | $\mathbf{0.72}_{\pm.01}$ | $\mathbf{0.76}_{\pm.00}$ | $\mathbf{0.93}_{\pm.03}$ | $\mathbf{0.92}_{\pm.00}$ |
| Disagree. | $\mathbf{0.37}_{\pm.01}$ | $\mathbf{0.54}_{\pm.01}$ | $\mathbf{0.61}_{\pm.06}$ | $\mathbf{0.80}_{\pm.01}$ | $\mathbf{0.37}_{\pm.03}$ | $\mathbf{0.46}_{\pm.01}$ | $\mathbf{0.63}_{\pm.05}$ | $\mathbf{0.75}_{\pm.01}$ | $0.44_{\pm.01}$ | $0.52_{\pm.01}$ | $0.69_{\pm.06}$ | $0.79_{\pm.01}$ |
| Random | $0.00_{\pm.00}$ | $0.00_{\pm.00}$ | $0.01_{\pm.05}$ | $0.02_{\pm.02}$ | $0.00_{\pm.00}$ | $0.00_{\pm.00}$ | $0.00_{\pm.04}$ | $0.02_{\pm.03}$ | $0.00_{\pm.00}$ | $0.00_{\pm.00}$ | $-0.05_{\pm.18}$ | $-0.04_{\pm.04}$ |
| Canonical | $0.01_{\pm.00}$ | $0.09_{\pm.00}$ | $0.28_{\pm.04}$ | $0.35_{\pm.01}$ | $0.00_{\pm.00}$ | $0.26_{\pm.01}$ | $0.24_{\pm.02}$ | $0.58_{\pm.01}$ | $0.00_{\pm.00}$ | $0.14_{\pm.00}$ | $0.42_{\pm.09}$ | $0.44_{\pm.00}$ |

conditional entropy $H(V_g|U_S)$ can be expressed as follows.

$$I(V_\mu, V_g|U_S) = \sum_{U_S} p(U_S) \sum_{V_\mu \in \{\pm 1\}} \sum_{V_g \in \{\pm 1\}} p(V_\mu, V_g|U_S) \log \left( \frac{p(V_\mu, V_g|U_S)}{p(V_\mu|U_S)p(V_g|U_S)} \right)$$

$$H(V_g|U_S) = -\sum_{U_S} p(U_S) \sum_{V_g \in \{\pm 1\}} p(V_g|U_S) \log(p(V_g|U_S))$$

Dividing $I(V_\mu, V_g|U_S)$ by $H(V_g|U_S)$ for normalization and restricting the size of $S$ for enabling computation, [18] defines the metric $\mathcal{K}(\mu)$ as:

$$\mathcal{K}(\mu) := \min_{U_S \text{ s.t.} |S| \leq 2} \frac{I(V_\mu, V_g|U_S)}{H(V_g|U_S)}. \tag{7}$$

**Results (Tables 15–16)** In Table 15, we show the average and standard deviation of the correlation scores of model-wise quantities (for one model per training procedure), following [18]. The model-wise inconsistency for model $\theta$ trained on $Z_n$ with procedure $P$ is $\mathbb{E}_{\Theta \sim \mathbf{\Theta}_{P|Z_n}} \mathbb{E}_X \text{KL}(f(\Theta, X)||f(\theta, X))$, and $Z_n$ was fixed here; similarly, model-wise disagreement is $\mathbb{E}_{\Theta \sim \mathbf{\Theta}_{P|Z_n}} \mathbb{E}_X \mathbb{I}[\, c(\Theta, X) \neq c(\theta, X) \,]$ where $c(\theta, x)$ is the classification decision of model $\theta$ on data point $x$. The average and standard deviation were computed over 4 independent runs that used 4 distinct subsamples of training sets as training data $Z_n$ and distinct random seeds for model parameter initialization, data mini-batching, and so forth.

Table 15 compares inconsistency and disagreement in terms of their correlations with the generalization gap (test loss minus training loss as defined in the main paper), test error, and test error minus training error. The training procedures analyzed here are the procedures that achieve low final randomness by either a vanishing learning rate or iterate averaging as in Figure 5. Tables (a), (b), and (c) differ in the models included in the analysis. As noted in Appendix C.1, since near-random models in the initial phase of training are not of practical interest, procedures with very high training loss were excluded from the analysis in the main paper. Similarly, Table 15-(b) excludes the models with high training loss using the same cut-off values as used in the main paper, and (c) does this with smaller (and therefore more aggressive) cut-off values (one half of (b)), and (a) does not exclude any model. Consequently, the average training loss is the highest in (a) and lowest in (c).

Let us first review Table 15-(a). The results show that inconsistency correlates well to generalization gap (test loss minus training loss) as suggested by our theorem, and disagreement correlates well to test error as suggested by the theorem of the original disagreement study [17]. Regarding 'test error minus training error' (last 4 rows): on CIFAR-10, training error is relatively small and so it approaches test error, which explains why disagreement correlates well to it; on the other datasets, 'test error minus training error' is more related to 'test loss minus training loss', which explains why inconsistency correlates well to it. The standard deviations are relatively small, and so the results are solid. (The standard deviation of $\Psi$ tends to be higher than the others for all quantities including the baseline 'Random', and this is due to the combination of the macro averaging-like nature of $\Psi$ and the smallness of $|\Pi_i|$ for some $i$'s, independent of the nature of inconsistency or disagreement.)

The overall trend of Table 15-(b) and (c) is similar to (a). That is, inconsistency correlates well to generalization gap (test loss minus training loss) while disagreement correlates well to test error, consistent with the results in the main paper and the original disagreement study. Comparing (a), (b), and (c), we also note that as we exclude the models with high training loss more aggressively (i.e., going from (a) to (c)), the correlation of disagreement to generalization gap (relative to that of inconsistency) improves and approaches that of inconsistency. For example, on CIFAR-100, the ratio of $\mathcal{K}$ for (disagreement, generalization gap) with respect to $\mathcal{K}$ for (inconsistency, generalization gap) improves from (a) 0.11/0.51=22% to (b) 0.26/0.47=55% to (c) 0.31/0.44=70%. With these models, empirically, high training loss roughly corresponds to the low confidence-level on unseen data, and so this observation is consistent with the theoretical insight that when the confidence-level on unseen data is high, disagreement should correlate to generalization gap as well as inconsistency, which is discussed in more detail in Appendix D.3.

Table 16 shows that the correlation of the estimate of $\mathcal{C}_P$ (defined in Section 2) to the generalization gap is generally as good as the estimate of $\mathcal{D}_P$ (defined in Theorem 2.1), which is consistent with the results in the main paper.

Table 16: Correlation analyses of the estimates of $\mathcal{C}_P$ and $\mathcal{D}_P$ (as defined in Section 2) using the rank-based and mutual information-based metrics from [18]. Training procedures with low final randomness of Figure 5. **Correlation scores**: see the caption of Table 15. **Target quantities**: Generalization gap (test loss minus training loss) analyzed in Theorem 2.1. **Observation**: The correlation of $\mathcal{C}_P$ to generalization gap is generally as good as $\mathcal{D}_P$, which is consistent with the results in the main paper.

| | CIFAR-10 | | | | CIFAR-100 | | | | ImageNet | | | |
| | MI-based | | Ranking | | MI-based | | Ranking | | MI-based | | Ranking | |
| | $\mathcal{K}$ | $\lvert S \rvert$=0 | $\Psi$ | $\tau$ | $\mathcal{K}$ | $\lvert S \rvert$=0 | $\Psi$ | $\tau$ | $\mathcal{K}$ | $\lvert S \rvert$=0 | $\Psi$ | $\tau$ |
|---|---|---|---|---|---|---|---|---|---|---|---|---|
| $\mathcal{C}_P$ | 0.38 | 0.59 | 0.71 | 0.83 | 0.49 | 0.53 | 0.80 | 0.80 | 0.75 | 0.75 | 0.59 | 0.92 |
| $\mathcal{D}_P$ | 0.38 | 0.59 | 0.80 | 0.84 | 0.42 | 0.47 | 0.75 | 0.76 | 0.81 | 0.81 | 0.60 | 0.94 |

## D.2  Training error

Tables 17 and 18 show the training error values (the average, minimum, median, and the maximum) associated with the empirical results reported in Section 3.1 and 3.2, respectively.

Table 17: Training error of the models analyzed in Section 3.1. The four numbers represent the average, minimum, median, and maximum values (%).

| | CIFAR-10 | | | | CIFAR-100 | | | | ImageNet | | | |
|---|---|---|---|---|---|---|---|---|---|---|---|---|
| Figure 1,3,6 | 0.4 | 0.0 | 0.0 | 8.8 | 4.7 | 0.0 | 0.1 | 53.6 | 24.9 | 3.0 | 23.5 | 56.6 |
| Figure 2 | 3.3 | 0.0 | 2.6 | 10.9 | 18.2 | 0.2 | 13.6 | 56.3 | 44.5 | 16.8 | 45.6 | 65.3 |
| Figure 5,7 | 0.5 | 0.0 | 0.0 | 10.1 | 4.0 | 0.0 | 0.0 | 54.5 | 16.3 | 0.1 | 11.3 | 59.0 |

Table 18: Training error of the models analyzed in Section 3.2. The four numbers represent the average, minimum, median, and maximum values (%).

| Case#1 | | | | Case#2 | | | | Case#3 | | | | Case#4 | | | | Case#5 | | | |
|---|---|---|---|---|---|---|---|---|---|---|---|---|---|---|---|---|---|---|---|
| 10.0 | 8.8 | 10.1 | 11.6 | 4.5 | 2.1 | 4.5 | 6.7 | 0.06 | 0.02 | 0.04 | 0.17 | 0.04 | 0.01 | 0.03 | 0.11 | 0.0 | 0.0 | 0.0 | 0.0 |
| Case#6 | | | | Case#7 | | | | Case#8 | | | | Case#9 | | | | Case#10 | | | |
| 0.7 | 0.3 | 0.8 | 1.1 | 0.1 | 0.1 | 0.1 | 0.2 | 4.8 | 3.9 | 4.9 | 5.5 | 2.7 | 1.9 | 2.9 | 3.3 | 2.8 | 0.5 | 1.8 | 6.5 |

## D.3  More on inconsistency and disagreement

Inconsistency takes how strongly the models disagree on each data point into account while disagreement ignores it. That is, the information disagreement receives on each data point is *binary* (whether the classification decisions of two models agree or disagree) while the information inconsistency receives is continuous and *more complex*. On the one hand, this means that inconsistency could use information ignored by disagreement and thus it could behave quite differently from disagreement as seen in our empirical study. On the other hand, it should be useful also to consider the situation where inconsistency and disagreement are highly correlated since in this case our theoretical results can be regarded as providing a theoretical basis for the correlation of not only inconsistency but also disagreement with generalization gap though indirectly.

To simplify the discussion towards this end, let us introduce a slight variation of Theorem 2.1, which uses 1-norm instead of the KL-divergence since disagreement is related to 1-norm as noted in Section 2.

**Proposition D.1** (1-norm variant of Theorem 2.1). *Using the notation of Section 2, define 1-norm inconsistency $\mathcal{C}_{1,P}$ and 1-norm instability $\mathcal{S}_{1,P}$ which use the squared 1-norm of the difference in place of the KL-divergence as follows.*

$$\mathcal{C}_{1,P} = \mathbb{E}_{Z_n}\mathbb{E}_{\Theta,\Theta'\sim\mathbf{\Theta}_{P|Z_n}}\mathbb{E}_X\|f(\Theta,X)-f(\Theta',X)\|_1^2 \qquad \text{(1-norm inconsistency)}$$

$$\mathcal{S}_{1,P} = \mathbb{E}_{Z_n,Z_n'}\mathbb{E}_X\|\bar{f}_{P|Z_n}(X)-\bar{f}_{P|Z_n'}(X)\|_1^2 \qquad \text{(1-norm instability)}$$

*Then with the same assumptions and definitions as in Theorem 2.1, we have*

$$\mathbb{E}_{Z_n}\mathbb{E}_{\Theta\sim\mathbf{\Theta}_{P|Z_n}}\left[\Phi_{\mathbf{Z}}(\Theta)-\Phi(\Theta,Z_n)\right] \le \inf_{\lambda>0}\left[\frac{\gamma^2}{2}\psi(\lambda)\lambda\left(\mathcal{C}_{1,P}+\mathcal{S}_{1,P}\right)+\frac{\mathcal{I}_P}{\lambda n}\right].$$

*Sketch of proof.* Using the triangle inequality of norms and Jensen's inequality, replace inequality (2) of the proof of Theorem 2.1

$$\mathbb{E}_{Z_n}\mathbb{E}_{\Theta \sim \mathbf{\Theta}_{P|Z_n}}\mathbb{E}_X \|f(\Theta, X) - \bar{f}_P(X)\|_1^2 \le 4\left(\mathcal{C}_P + \mathcal{S}_P\right) \tag{2}$$

with the following,

$$\mathbb{E}_{Z_n}\mathbb{E}_{\Theta \sim \mathbf{\Theta}_{P|Z_n}}\mathbb{E}_X \|f(\Theta, X) - \bar{f}_P(X)\|_1^2$$
$$\le 2\mathbb{E}_{Z_n}\mathbb{E}_{\Theta \sim \mathbf{\Theta}_{P|Z_n}}\mathbb{E}_X \left[\|f(\Theta, X) - \bar{f}_{P|Z_n}(X)\|_1^2 + \|\bar{f}_{P|Z_n}(X) - \bar{f}_P(X)\|_1^2\right]$$
$$\le 2\mathbb{E}_{Z_n}\mathbb{E}_{\Theta,\Theta' \sim \mathbf{\Theta}_{P|Z_n}}\mathbb{E}_X \|f(\Theta, X) - f(\Theta', X)\|_1^2 + 2\mathbb{E}_{Z_n}\mathbb{E}_{Z_n'}\mathbb{E}_X \|\bar{f}_{P|Z_n}(X) - \bar{f}_{P|Z_n'}(X)\|_1^2$$
$$= 2\left(\mathcal{C}_{1,P} + \mathcal{S}_{1,P}\right).$$

$\square$

Now suppose that with the models of interest, the confidence level of model outputs is always high so that the highest probability estimate is always near 1, i.e., for any model $\theta$ of interest, we have $1 - \max_i f(x, \theta)[i] < \epsilon$ for a positive constant $\epsilon$ such that $\epsilon \approx 0$ on any unseen data point $x$. Let $c(\theta, x)$ be the classification decision of $\theta$ on data point $x$ as in Section 2: $c(\theta, x) = \arg\max_i f(\theta, x)[i]$. Then it is easy to show that we have

$$\frac{1}{2}\|f(\theta, x) - f(\theta', x)\|_1 \approx \mathbb{I}\left[\, c(\theta, x) \ne c(\theta', x)\,\right]. \tag{8}$$

Disagreement measured for shared training data can be expressed as

$$\mathbb{E}_{Z_n}\mathbb{E}_{\Theta,\Theta' \sim \mathbf{\Theta}_{P|Z_n}}\mathbb{E}_X \mathbb{I}\left[\, c(\Theta, X) \ne c(\Theta', X)\,\right]. \tag{9}$$

Comparing (9) with the definition of $\mathcal{C}_{1,P}$ above and considering (8), it is clear that under this high-confidence condition, disagreement (9) and 1-norm inconsistency $\mathcal{C}_{1,P}$ should be highly correlated; therefore, under this condition, Proposition D.1 suggests the relation of disagreement (measured for shared training data) to generalization gap indirectly through $\mathcal{C}_{1,P}$.

While this paper focused on the KL-divergence-based inconsistency motivated by the use of the KL-divergence by the existing algorithm for encouraging consistency, the proposition above suggests that 1-norm-based inconsistency might also be useful. We have conducted limited experiments in this regard and observed mixed results. In the settings of Appendix D.1, the correlation scores of 1-norm inconsistency with respect to generalization gap are generally either similar or slightly better, which is promising. As for consistency encouragement during training, we have not seen a clear advantage of using 1-norm inconsistency penalty over using the KL-divergence inconsistency penalty as is done in this paper, and more experiments would be required to understand its advantage/disadvantage.

In our view, however, for the purpose of encouraging consistency during training, KL-divergence inconsistency studied in this paper is more desirable than 1-norm inconsistency in at least three ways. First, minimization of the KL-divergence inconsistency penalty is equivalent to minimization of the standard cross-entropy loss with soft labels provided by the other model; therefore, with the KL-divergence penalty, the training objective can be regarded as a weighted average of two cross-entropy loss terms, which are in the same range (while 1-norm inconsistency is not). This makes tuning of the weight for the penalty more intuitive and easier. Second, optimization of the standard cross-entropy loss term with the KL-divergence inconsistency penalty has an interpretation of functional gradient optimization, as shown in [19]. The last (but not least) point is that optimization may be easier with KL-divergence inconsistency, which is smooth, than with 1-norm inconsistency, which is not smooth.

Related to the last point, disagreement, which involves $\arg\max$ in the definition, cannot be easily integrated into the training objective, and this is a crucial difference between inconsistency and disagreement from the algorithmic viewpoint.

Finally, we believe that for improving deep neural network training, it is useful to study the connection between generalization and discrepancies of model outputs in general including instability, inconsistency, and disagreement, and we hope that this work contributes to progress in this direction.

