# OpenReview forum: "Inconsistency, Instability, and Generalization Gap of Deep Neural Network Training"
_NeurIPS.cc/2023/Conference — NeurIPS 2023 poster_

### Official Review · Reviewer_TByp · 2023-07-04

**Soundness:** 3 good
**Presentation:** 3 good
**Contribution:** 2 fair
**Rating:** 5
**Confidence:** 4

**Summary:**

This manuscript propose new notions of inconsistency, instability, and information-theoretifc instability based on the output confidence score to estimate the generalization gap of deep neural networks. Theoretical and empirical results are presented and show that the proposed notions, especially inconsistency, are correlated well with well-trained neural networks.

**Strengths:**

1. Three novel notions are proposed to measure the generalization gap of neural networks;

2. Extensive experiments have been conducted to verfify the good correlation of instability and inconsistency with the generalization gap of neural networks;

3. The manuscript is well organized and the writing is clear;

4. Emprical study show the better correlation of the inconsistency than disagreement.

**Weaknesses:**

1. The novelty of proposed measurements are limited:
    - The Inconsistency and Instability are very similar to the definition of disagreement, while the former notions replace the outout from one-hot predictions to softmax confidence score.
    -  The Instabilty of model parameter distributions ($\mathcal{I}_P$) can be regarded as a kind of algorithm stability. Therefore, I suggest the authors to provide a discussion on the differentce or advantages w.r.t. former definition of algorithm stabiliy.

2. Marginal contributions on the theoretical results (Theorem 2.1):
    - The upper bound in Theorem 2.1 is not a uniform convergence-based generalization bound, and does not show the relation to training sample size and may be loose, which greatly undermines the significance of the theoretical results;
    - The right hand of the given upper bound is hard to estimate due to the existence of the Instabiliyt of model parameter distributions $\mathcal{I}_P$, although In consistency $\mathcal{C}_P$ and Instability $\mathcal{S}_P$ can be convenient to estimate on unlabeled data.


Based on above weaknesses and considering the marginal contributions on the proposed notions and theoretical results, I think this work is slightly below the acceptance bar.

**Questions:**

See weaknesses.

**Limitations:**

See weaknesses.

---

> ### Author Rebuttal · Authors · 2023-08-08
>
> > The Inconsistency and Instability are very similar to the definition of disagreement, while the former notions replace the outout from one-hot predictions to softmax confidence score.
>
> We find it interesting that in spite of the similarity in the definitions, they behave quite differently, as shown in the submission as well as the one-page pdf attached to the global response.  The theoretical justifications are also quite different -- the disagreement study shows that disagreement=test error (not the loss gap like our theorem) if the training procedure produces well-calibrated ensembles.  At a high level, we believe that it would be useful to study the connection between generalization performance and discrepancies of model outputs in general (whether instability, inconsistency, disagreement, or else), and considering the clear observable differences from disagreement, we feel that it is useful to study instability/inconsistency as well as disagreement.
>
> > The Instabilty of model parameter distributions (${\mathcal I_P}$) can be regarded as a kind of algorithm stability. Therefore, I suggest the authors to provide a discussion on the differentce or advantages w.r.t. former definition of algorithm stabiliy.
>
> In our view, ${\mathcal I_P}$ is more like the covering number or entropy than algorithmic stability.  ${\mathcal I_P}$ appears in recently popularized information theoretical analyses, which we used to quantify randomness effects of learning algorithms into our generalization bound.  We will add a discussion in our revision.
>
> > The upper bound in Theorem 2.1 is not a uniform convergence-based generalization bound, and does not show the relation to training sample size and may be loose, which greatly undermines the significance of the theoretical results
>
> In the theorem, $n$ is the training sample size.
>
> Information theoretical results using mutual information inherently hold in expectation because mutual information is defined with respect to expectation. The results can be easily translated into a uniform convergence bound if we choose a prior and use the KL divergence with respect to the prior (instead of mutual information), and we thought this was well known. We will add a discussion.
>
> > The right hand of the given upper bound is hard to estimate due to the existence of the Instabiliyt of model parameter distributions, although Inconsistency and Instability can be convenient to estimate on unlabeled data.
>
> The estimation of ${\mathcal I_P}$ is possible using bootstrap, though not easy as noted, and we are happy to add a discussion in this regard.  On the other hand, ${\mathcal I_P}$ is a standard quantity in the mutual information-based analyses, and so all the previous analyses using this quantity share the same trait.  While we employ ${\mathcal I_P}$ to deal with the stochastic nature of neural network training, we would like to emphasize that the strength of this analysis is the inclusion of new easily-measurable quantities, as noted, and that the main focus of this paper is the empirical study of these new quantities.

---

### Official Review · Reviewer_3GRr · 2023-07-05

**Soundness:** 3 good
**Presentation:** 3 good
**Contribution:** 2 fair
**Rating:** 6
**Confidence:** 4

**Summary:**

The paper presents two measures for a stochastic training algorithm: inconsistency and instability. The former measures the inconsistency (or "disagreement") within the random ensemble of models trained from the same training set. The latter measures the inconsistency of two ensembled predictors, each obtained from an ensemble trained from an independent training set. A theorem is presented stating that the sum of inconsistency and instability modulates the mutual information (between the training set and algorithm output) in the generalization bound of Xu/Raginsky' 2017.  Empirical investigation is performed to assess the predictiveness of inconsistency and instability for generalization gap. Algorithmic implications are also investigated.

**Strengths:**

To this reviewer, a particularly novel and interesting aspect of this work is bringing the notion of inconsistency, or "within-training-set agreement" into the landscape of generalization bounds. This notion is akin to the notion of "generalization disagreement equality" (GDE) in the work of Jiang et al 2022 (reference [17] of this paper).  Notably -- although not adequately discussed by the authors -- a sufficient condition of GDE is a notion of calibration in [17]. A potential impact of this work is extending the development of information-theoretic generalization bounds to include the calibration-alike quantities. This, I found intriguing and inspiring.

The theoretical development is light. Nonetheless interesting.

Among various empirical results, the most interesting and novel aspect to this reviewer is the observation that inconsistency is more predictive for generalization than sharpness. Algorithmic exploitation of this aspect is also interesting.

**Weaknesses:**

Some conclusions from the empirical study appear speculative to this reviewer. For example, the authors hypothesized that a low degree of randomness is required for inconsistency+instability to be predictive, and in other places the author attributed complex phenomena arising from experiments to the interaction with the mutual information term. It is desirable that such claims are better corroborated.

**Questions:**

1. Are the notions of inconsistency and instability related to the notion of functional-CMI (or functional MI) in the work of Harutyunyan et al, "Information-theoretic generalization bounds for black-box learning algorithms", NeurIPS 2021? Exploring this connection might enhance this work.

2. What is the loss function used in evaluating generalization gaps in the experiments?



**Limitations:**

Nothing to add.

---

> ### Author Rebuttal · Authors · 2023-08-08
>
> > Are the notions of inconsistency and instability related to the notion of functional-CMI (or functional MI) in the work of Harutyunyan et al, "Information-theoretic generalization bounds for black-box learning algorithms", NeurIPS 2021? Exploring this connection might enhance this work.
>
> It seems to us that the notions of inconsistency and instability are not the same as the notion of functional-CMI (or functional MI) because the latter does not lead to a Bernstein-style bound and does not use variance information. The latter might lead to the so-called first-order bound, but we need to study it further to understand the relationship.  We will be happy to add a discussion.
>
> > What is the loss function used in evaluating generalization gaps in the experiments?
>
> It was the cross-entropy loss.

---

> > ### Comment · Reviewer_3GRr · 2023-08-22
> > **Thank you for the reply.**
> >
> > I will keep the rating.

---

### Official Review · Reviewer_k85B · 2023-07-06

**Soundness:** 4 excellent
**Presentation:** 4 excellent
**Contribution:** 3 good
**Rating:** 4
**Confidence:** 4

**Summary:**

This paper investigates the generalization gap in deep neural networks, and propose that this gap is influenced by the inconsistency and instability of model outputs, two quantities which are defined by the authors, and justified theoretically via a new information theoretic generalization bound. The authors conduct empirical studies that confirm the predictive power of inconsistency and instability on the generalization gap, and demonstrate that explicitly reducing inconsistency during training improves performance.

**Strengths:**

The paper is generally well-written, and includes several interesting observations. While I'm unsure of the novelty of Theorem 2.1, its form is compelling and I like the fact that $\mathcal{D}_P$ and $\mathcal{C}_P$ can be estimated efficiently (unlike the mutual information term $\mathcal{I}_P$ that also appears in other information theoretic bounds). I also find the results at the end on explicitly encouraging consistency interesting and a strong contribution -- in my opinion it would be good to expand on this aspect of the paper.

**Weaknesses:**

While the form of the bound in Theorem 2.1 is compelling, I have some concerns with its novelty/improvement relative to existing results (see questions below). In particular, it's unclear to me that the bound represents an improvement on existing information theoretic generalization bounds in the literature.

On the empirical side, I think a more rigorous correlation analysis of the $\mathcal{C}_P/\mathcal{D}_P$ metrics along the lines of prior work (e.g. Jiang et al. 2020) would help strengthen the claims of the paper, since the empirical correlation between these measures and the generalization gap seems to be one of its main contributions. I also think it would be helpful to have a more detailed comparison of the inconsistency/error relationship compares with the disagreement/error relationship observed in prior work, as these seem very closely related.

Overall, while the paper is well-written and contains some interesting insights, at the current stage I think it lacks sufficient novelty/improvement on the theoretical side, and comparison to prior work on the empirical side to recommend acceptance.

**References**
Yiding Jiang, Behnam Neyshabur, Hossein Mobahi, Dilip Krishnan, Samy Bengio, Fantastic Generalization Measures and Where to Find Them, 2020.

**Questions:**

- How does the bound Theorem 2.1 compare with the large existing literature of information-theoretic generalization bounds? Perhaps the simplest of these from Xu & Raginsky (2017) is of the form $\sqrt{2\sigma^2 I/n}$, upon which your result seems like a small improvement at best, and only when $\mathcal{D}_P$ is very small. More recent bounds (e.g. Steinke and Zakynthinou, 2020 using the conditional mutual information) have made significant improvements on this, and so clarification as to which regimes your result provides an improvement would be helpful.
- Relatedly, is there any evidence that $\mathcal{D}_p$ is the dominant term in Theorem 2.1? Previous work has noted that mutual-information based bounds can be extremely large (though they are difficult to numerically estimate), and seemingly here the term $\mathcal{I}_P$ would dominate.
- Could the authors clarify what is being varied in the plots in, e.g. Figures 1 and 2? Are these plots illustrating the generalization gap/$\mathcal{D}_P$ in time, i.e. as a function of the iteration of optimization? If so, it would be useful if there was some indication of the direction of time in this plot.
- Do the authors have any hypotheses for why the disagreement/error relationship exhibits different behavior than the generalization gap/inconsistency relationship (maybe specifically in the zero training error regime, in which the generalization gap = test error)? It seems like this may have to do with the distribution of the logits in the trained models (which is ignored by the disagreement but not by the inconsistency metric), which would be very interesting to understand.
- Could the authors explicitly state the penalty that is added to the loss function during training to encourage consistency and explain how it is computed?

**References**
Thomas Steinke and Lydia Zakynthinou, Reasoning About Generalization via Conditional Mutual Information, 2020.

---

> ### Author Rebuttal · Authors · 2023-08-08
>
> Regarding theoretical novelty: in our view, our analysis is quite different from Xu & Raginsky (2017) as their bound is a sub-Gaussian bound which cannot lead to a faster rate than $\sqrt{1/n}$ as long as the noise variance is nonzero. In comparison we have a Bernstein-style bound, which is needed for a faster rate in classical statistics and quite different from a sub-Gaussian bound.  Also our bound incorporates new quantities (inconsistency and instability of model outputs) different from the noise variance. Steinke and Zakynthinou (2020), using the conditional mutual information, is orthogonal and complementary to our work. We focus on new easily-measurable quantities and derive a new Bernstein bound to utilize them, but it will be interesting to incorporate conditional mutual information into the analysis in the future.
>
> > a more rigorous correlation analysis of the metrics along the lines of prior work (e.g. Jiang et al. 2020) would help strengthen the claims of the paper
>
> As suggested, we performed correlation analyses using the metrics from [Jiang et al. 2020] and confirmed that the results are consistent with the submission.  We included a part of the results (comparison between inconsistency and disagreement) in the one-page pdf attached to the global response.  We will include the complete results in the paper (including other settings and the inconsistency+instability results).  It would indeed improve the paper.  We appreciate the suggestion.
>
> > I also think it would be helpful to have a more detailed comparison of the inconsistency/error relationship compares with the disagreement/error relationship observed in prior work, as these seem very closely related.
>
> We performed this comparison using the metrics from [Jiang et al. 2020] and included part of the results in the one-page pdf (global response).  The results show that essentially inconsistency is correlated to generalization gap while disagreement is correlated to test error.  This is consistent with our submission and the previous study -- the inconsistency/gap relation is suggested by our theorem and the disagreement/error relation is suggested by the theorem of the disagreement paper (disagreement = test error if the training procedure produces well-calibrated ensembles).
>
> > what is being varied in the plots in, e.g. Figures 1 and 2?
>
> In Figures 1--4, the learning rate and training length were varied, and the points are connected in the order of training length.  This was said somewhere in the text, but now we realize that it should be said also in the captions and there should be arrows in the graphs, as suggested.  We will improve it if accepted.
>
> > Do the authors have any hypotheses for why the disagreement/error relationship exhibits different behavior than the generalization gap/inconsistency relationship (maybe specifically in the zero training error regime, in which the generalization gap = test error)? It seems like this may have to do with the distribution of the logits in the trained models (which is ignored by the disagreement but not by the inconsistency metric), which would be very interesting to understand.
>
> As noted, inconsistency takes how strongly the models disagree into account while the disagreement ignores it.  Suppose that the confidence level of model outputs (on unseen data) is fixed to a high value so that the models always either strongly agree or strongly disagree.  In this situation, inconsistency and disagreement should be highly correlated.  Further assume that test error and generalization gap are highly correlated, and then the disagreement/error relationship and the inconsistency/gap relationship should be similar.  However, in the reality, even if all the models have zero training error, their confidence level on unseen data could vary, thus, the strength of agreement between models would not be binary, which means that inconsistency could use information ignored by disagreement.  This is an interesting point, and we will add a discussion to the revision.
>
> > Could the authors explicitly state the penalty that is added to the loss function during training to encourage consistency and explain how it is computed?
>
> For encouraging consistency, two models are trained simultaneously with two distinct random sequences (for initialization, mini-batch sampling, etc.) with the inconsistency penalty term, which is the KL-divergence of the model output with respect to the model output of the other model.  With the absence of unlabeled data, Algorithm 1 (in the Appendix) was used, and with unlabeled data (Table 4), Algorithm 2, which computes the inconsistency penalty term on the unlabeled data, was used.

---

### Official Review · Reviewer_ARdH · 2023-07-07

**Soundness:** 3 good
**Presentation:** 3 good
**Contribution:** 3 good
**Rating:** 6
**Confidence:** 3

**Summary:**

In this work, the authors introduce the ideas of “instability” and “inconsistency” of model outputs, and investigate the relationship between these quantities and the generalization gap. In particular, they empirically find a positive correlation between instability + inconsistency and the generalization gap. They further show that inconsistency can be more predictive of the generalization gap than m-flatness.

**Strengths:**

1. The paper is written cleary, well–organized, and well-motivated. The authors do a good job in providing some intuition behind the mathematical definitions of inconsistency and instability.
2. The paper includes extensive experiments to support their hypothesis (e.g., use of various architectures, vision and text datasets, training methods, etc.)


**Weaknesses:**

1. In Figure 2, the authors observe that $\mathcal{D_P}$'s predictive ability of generalization gap is sensitive to the learning rate. The authors further suggest that when the “final randomness is high,”  $\mathcal{D_P}$’s predictive ability of the generalization gap is not as strong. In practice, practitioners may use a large learning rate and small batch size to obtain well-generalizing models (and so the final randomness would be high in this situation). Thus, $\mathcal{D_P}$ may not be useful here. (In some sense, it is not surprising that low final randomness correlates with $\mathcal{D}_{P}$’s predictive ability of generalization gap).
2. The authors choose a constant learning rate schedule for the starting experiments to avoid confounding variables. However, the architectures used in the experiments have normalization layers, which can induce learning rate schedules. Perhaps running the preliminary experiments with at least one architecture w/out any normalization layers would be beneficial.
3. The benefit of $\mathcal{D}_{P}$ over prior metrics such as disagreement is not evident. In [17], test error is calculated at the end of training (when the train error is nearly zero). Since this is not true in figure 10, I would be interested in a plot of the gap between train accuracy and test accuracy  vs. training loss instead of figure 10d. (aside: the gap between train accuracy and test accuracy is an alternate definition of generalization gap to the definition used in this work). Perhaps this would lead to a more appropriate comparison of inconsistency vs. disagreement detailed in lines 113-127.
More generally, it would be nice to also include plots for this alternative definition of generalization gap, especially since experiment performance in the paper is often measured via test error.
4. In the appendix (lines 600-604), $\rho$ was set based on reference or the development data. However, the optimal value of $\rho$ can be quite sensitive to the architecture used. Thus, it would be nice to use some type of grid search to choose $\rho$ for fair comparison in e.g., figure 8.


**Questions:**

It would be helpful to include legends in each figure (e.g., figure 4 is missing a legend. I assume the legend is consistent with figure 2. However, it would be convenient to include the legend again.)

What are the training errors for models in each experiment?


**Limitations:**

The authors have described the limitations.

---

> ### Author Rebuttal · Authors · 2023-08-08
>
> > In practice, practitioners may use a large learning rate and small batch size to obtain well-generalizing models (and so the final randomness would be high in this situation). Thus, ${\mathcal D_P}$ may not be useful here.
>
> Even when the initial learning rate is large, the final randomness can be reduced by either decaying the learning rate (e.g., by multiplying 0.1 a few times, using the cosine schedule, etc.) or performing iterate averaging, and both are commonly practiced.  Also, Section 3.2 experiments with known high-performing training procedures and shows that inconsistency is predictive of generalization gap.  And so we believe that inconsistency can be useful in practical settings.
>
> > The authors choose a constant learning rate schedule for the starting experiments to avoid confounding variables. However, the architectures used in the experiments have normalization layers, which can induce learning rate schedules. Perhaps running the preliminary experiments with at least one architecture w/out any normalization layers would be beneficial.
>
> Thanks for the suggestion. In our preliminary experiments, we tried some normalization-free network architectures and observed that inconsistency is predictive of generalization gap.  We designed the main experiments without them as their training was too expensive.  However, we will give them another try.
>
> > I would be interested in a plot of the gap between train accuracy and test accuracy vs. training loss instead of figure 10d.
>
> The suggested plot turned out to be similar to 10a (the gap between training loss and test loss vs. training loss) in this particular setting of Fig 10, but it may be useful in some other settings.
>
> > More generally, it would be nice to also include plots for this alternative definition of generalization gap, especially since experiment performance in the paper is often measured via test error.
>
> We conducted additional correlation analyses, which include this alternative definition of gap (test error minus training error), and showed some results in the one-page pdf attached to the global response.  We hope that the analyses of this form, when added to the paper, will clarify the interesting difference from (and the merit over) disagreement.
>
> > In the appendix (lines 600-604), $\rho$ was set based on reference or the development data. However, the optimal value of $\rho$ can be quite sensitive to the architecture used. Thus, it would be nice to use some type of grid search to choose for fair comparison in e.g., figure 8.
>
> Lines 600-604 describes selection of $\rho$ for training SAM models, and yes, $\rho$ was set to the optimal value for each combination of the architecture and data.  When the optimal value is known from the literature, we used that value but also made sure it was actually optimal by testing the grid in the neighbor.  When the optimal value was not known for the combination of the architecture and data, we performed a grid search (on a held-out subsample of training data serving as development data).
>
> As for measuring 1-sharpness, we tested the values on a grid in the neighbor of the $\rho$ value used for training SAM (= the optimum $\rho$ for SAM) and found that the prediction power of 1-sharpness is similar in this neighbor.  Therefore, for simplicity, we used this optimum $\rho$ for SAM also for 1-sharpness for producing the results in the paper, as described in Line 624.
>
> > It would be helpful to include legends in each figure (e.g., figure 4 is missing a legend. I assume the legend is consistent with figure 2. However, it would be convenient to include the legend again.)
>
> To save space, the legend is sometimes described in the caption, e.g., "Same procedures and legend as in Fig 2" in the caption of Fig 4, but we will improve it if accepted.
>
> > What are the training errors for models in each experiment?
>
> The training error varies and we cannot describe it for all the experiments here, but we will include it in the paper if accepted.  Essentially, it is often near zero for CIFAR-10 (easier data; tiny images with only 10 classes) and for harder data (e.g., ImageNet; much larger images with 1000 classes), it varies more.

---

> > ### Comment · Reviewer_ARdH · 2023-08-21
> >
> > I acknowledge and appreciate the authors' responses. I intend to keep my score.

---

### Author Rebuttal · Authors · 2023-08-08

Thank you very much for the valuable feedback.

In response to the suggestions, we conducted additional correlation analyses using the metrics from [Jiang et al. 2020], with generalization gap (loss difference), test error, and test error minus training error.  One of the metrics from [Jiang et al. 2020] is more rigorous in the sense that it was designed to eliminate spurious correlations.  The results are consistent with the submission.  We uploaded a one-page pdf that contains a part of the results.  We will include the complete results in the paper, if accepted, which would be a great addition.  We really appreciate the helpful suggestions.

The rest of our response is to each reviewer individually.

---

### Decision · Program_Chairs · 2023-09-21

**Decision:**

Accept (poster)

**Comment:**

The paper presents an elegant framework for relevant know-hows in generalization of deep learning. The notion of inconsistency seamlessly interprets the generalization power of ensemble models while the notion of instability nicely complements the notion of stability in generalization of deep learning. I am mostly excited by the elegance of the framework and its relevance to the disagreement framework that was recently developed to explain the generalization of deep nets. While there are still many concerns for this framework, both practical and theoretical, I hope that this work might be able to bring more clarity to generalization of deep nets.